# Consumer-driven nutrient recycling of freshwater decapods: Linking ecological theories and application in integrated multitrophic aquaculture

Gabriela Musin[1], María Victoria Torres[2,3], Débora de Azevedo Carvalho[1]*

1 Instituto Nacional de Limnología, CONICET and Universidad Nacional del Litoral, Santa Fe, Argentina, 2 Centro de Investigaciones Científicas y Transferencia Tecnológica a la Producción, CONICET, Diamante, Entre Ríos, Argentina, 3 Facultad de Ciencia y Técnica, Universidad Autónoma de Entre Ríos, Entre Ríos, Argentina

☯ These authors contributed equally to this work.
* dazevedo@inali.unl.edu.ar

**Data Availability Statement:** All files are available from the https://ri.conicet.gov.ar/handle/11336/162063.

## Abstract

The Metabolic Theory of Ecology (MTE) and the Ecological Stoichiometry Theory (EST) are central and complementary in the consumer-driven recycling conceptual basis. The understanding of physiological processes of organisms is essential to explore and predict nutrient recycling behavior, and to design integrated productive systems that efficiently use the nutrient inputs through an adjusted mass balance. We fed with fish-feed three species of decapods (prawn, anomuran, crab) from different families and with aquacultural potential to explore the animal-mediated nutrient dynamic and its applicability in productive systems. We tested whether body mass, body elemental content, and feeds predict N and P excretion rates and ratios within taxa. We also verified if body content scales allometrically with body mass within taxa. Finally, we compared the nutrient excretion rates and body elemental content among taxa. N excretion rates of prawns and anomurans were negatively related to body mass, emphasizing the importance of MTE. Feed interacted with body mass to explain P excretion of anomurans and N excretion of crabs. Body C:N content positively scaled with body mass in prawns and crabs. Among taxa, prawns mineralised more N and N:P, and less P, and exhibited higher N and C body content (and lower C:N) than the other decapods. Body P and N:P content were different among all species. Body content and body mass were the main factors that explained the differences among taxa and influence the role of crustaceans as nutrient recyclers. These features should be considered to select complementary species that efficiently use feed resources. Prawns need more protein in feed and might be integrated with fish of higher N-requirements, in contrast to crabs and anomurans. Our study contributed to the background of MTE and EST through empirical data obtained from decapods and it provided insightful information to achieve more efficient aquaculture integration systems.

**Funding:** Fondo para la Investigación Científica y Tecnológica, Grant/Award Number: PICT 2016-2542 and PICT 2018-00690 funded this study. CONICET supported the postdoctoral fellowship of G.M. and M.V.T. The funders had no role in study design, data collection and analysis, decision to publish, or preparation of the manuscript.

**Competing interests:** The authors have declared that no competing interests exist.

# Introduction

Consumers are important nutrient recyclers in aquatic ecosystems as a source or sink for elements such as carbon (C), nitrogen (N) and phosphorus (P) [1,2]. The excretion and egestion of waste products are immediate processes by which animals can be a source of nutrients for primary producers and heterotrophic microorganisms [3]. Animals also constitute nutrient pools, as they grow and reproduce [1,2,4]. As stated by [5], "what animals eat and excrete shapes their role in ecological communities and determines their contribution to the flux of energy and materials in ecosystems". In this sense, two ecological theories are central and complementary in the consumer-driven recycling conceptual basis [6]: the Metabolic Theory of Ecology (MTE) [7] and the Ecological Stoichiometry Theory (EST) [8]. Whereas one emphasizes on energy (MTE), the other does so on matter (EST) [5].

The MTE states that the rate at which organisms take up, transform, and expend energy and materials is the most fundamental biological rate. Organisms are influenced by intrinsic (body size–hereafter, body mass–and stoichiometry of organisms) and extrinsic (temperature) factors, which, in turn, obey chemical and physical principles [7]. According to the EST, the organismal response to food composition is a useful tool to predict the stoichiometric homeostasis of a given consumer through the selective retention and release (excretion and egestion) of elements like C, N and P. Rates and ratios by which animals recycle nutrients reflect the element imbalance between their bodies and their food [8]. The mass-specific excretion rate of nutrients (i.e., nutrients excreted per unit of body mass per unit time) generally decrease with increasing body mass due to allometric restrictions in metabolism [3,8–10]. A central concept in EST is the Growth Rate Hypothesis (GRH) [8,11], which theorizes that differential allocation of body P content results from differential allocation of P-rich RNA required for protein synthesis during growth. Organisms with high rates of protein synthesis might have lower N:P content. The GRH also predicts that body P content should be high in small-bodied organisms because these organisms tend to have faster growth rates.

The N:P released by a homeostatic consumer increases with food N:P and decreases with body N:P content [12]. Studies found mixed results in aquatic vertebrates and invertebrates, both supporting and contradicting this mass balance model [13–17]. The variable accuracy of diet and body content as predictors of N:P excretion has been attributed to the flexible homeostasis of organisms [15,16], taxa-specific mechanisms [17], biotic and abiotic factors [18], and resource quality [19]. These mixed results must be due to taxa-specific metabolic processes. Body mass and taxonomic identity mainly predict excretion rates [20], but a better understanding of taxa-specific metabolic processes are necessary to predict animal-mediated nutrient recycling. Incubating animals in containers with a known volume of water is a common procedure to measure nutrient mineralisation rates [13,21]. However, time and conditions of incubation could also affect rate values and, therefore, experimental design should consider it [22].

It is hypothesized that variation in body stoichiometry of organisms is driven by taxon-specific constraints imposed by phylogeny [23–26], differential growth rates and allometry [8,11], structural differences in material allocation [8,27,28], and trophic position [23,25,29]. In this last case, trophic position should predict body composition because organisms should minimize imbalances between the elemental body content and their diet. In this way, predators might have low C:nutrient and high N:P content because they have adapted to ingest high-nutrient food better than lower trophic-positioned organisms [24]. However, high C:nutrient could be found in organisms with C-rich structures, such as the chitin of the exoskeleton, due to differential nutrient allocation [8,27]. In addition, variation in body composition might be

due to ontogenetic changes in body stoichiometry and/or P demand associated with faster growth rates [28,30].

In lowland rivers in the Atlantic slope of southern South America, five families of decapods (Sergestidae, Palaemonidae, Parastacidae, Aeglidae and Trichodactylidae) comprise the littoral-benthic communities of freshwater systems [31–34]. Despite the phylogenetic distance [35–38], differences in the ventrally folded pleon (carcinization) [39] and lifestyle [33], the trophic habit of these decapods is mostly omnivorous, generalist and opportunistic [32,34,40]. They fulfill key functions in the ecological processes of natural environments by crushing and processing decomposing plant material and by consuming aquatic invertebrates [40–44]. Although their trophic habit reflects the importance of vegetal and animal components, their diet could vary during ontogeny and interspecifically, and in response to extrinsic factors [45–47]. The trophic habit of these organisms is an advantage in captivity conditions because it allows an acceptance of artificial feed and enables experimental aquaculture research [48,49].

Nowadays, the accumulated information about these decapods is substantial. They display different morphological and physiological traits among families, such as foregut types [43,50,51], predation strategies [52–54], enzyme and metabolic activity [49,55,56], and oxygen consumption [57], which certainly influence resource acquisition, assimilation, and excretion. The holistic overview of this existing information, in addition to an experiment in a laboratory-controlled environment, could be a useful way to understand the taxa-specific differences in stoichiometry and metabolic rates (e.g., excretion) that influence the nutrient dynamic of natural and artificial aquatic systems.

There is an increasing interest in cultivating native species adapted to local conditions to conserve local biodiversity [58–60]. The use of native crustaceans in aquaculture systems is growing in South America [61–65]. The potential of using them as nutrient recycling organisms in fish production systems is high if we consider their wide trophic habits, their great acceptance of artificial feed, and the multiple uses of this by-product to add revenue to the production [48,66,67]. This approach is used to deal with the environmental burden of intensive farming and is coupled with the practice of integrated multitrophic aquaculture (IMTA) [68]. The low diversity of these artificial aquatic systems implies a strong influence of few taxa on the nutrient turnover [14,69] and emphasizes the importance of studying species with aquacultural potential and interest.

Coupling experimental aquaculture research with ecological theories is interesting to explore the animal-mediated nutrient dynamic and its applicability in productive systems. Here, we tested how body mass, body elemental content and feed explain nutrient recycling and discuss the results from an ecological and applied point of view. We used three species of decapods from different families, with antecedents about trophic and digestive ecology and with aquacultural potential use. We explored their potential role as nutrient recyclers in natural and artificial systems, such as in an IMTA, through experimental research using two commercial feeds elaborated for detritivorous and omnivorous fish.

In this context, we asked three main questions and suggested their corresponding hypotheses and predictions. **1**- Could body mass, body elemental content and type of feed predict N and P excretion rates and ratio within taxa? **1a**- Body mass is the variable that best explains the variation in N and P excretion, according to MTE. **1b**- N and P excretion rates negatively scale with body mass. **1c**- N:P excretion rates decrease with increasing N:P body content, as previously postulated [12]. **2**- Does body elemental content scale allometrically (with body mass) within taxa? **2a**- Body P content is negatively related to body mass, according to GRH. **3**- Are there differences with respect to the excretion rate and body elemental content among taxa? **3a**- Body C:N content increases in carcinized decapods and decreases in higher trophic positioned species, while body N:P increases in more carnivorous species.

## Material and methods

### Studied organisms

Decapod species used in this experiment represent three different families of neotropical decapod crustaceans. *Macrobrachium borellii* is a prawn of the Palaemonidae family, with wide distribution in La Plata Basin of northern Argentina, Paraguay and southern Brazil [70,71]. Its natural diet exhibits a significant presence of animal items such as dipterans and oligochaeta larvae, and a low importance of vegetal remains and algae [33,40,44]. It is a species characterized by moving in the water column and towards the littoral vegetation [50,72]. *Aegla uruguayana* belongs to the Aeglidae family, which has a unique genus endemic to southern South America. This species presents benthic habits, typically sheltered at the bottom and under rocks of current rivers and streams [70,73], and displays strong swimming habits [74,75]. Its natural diet consists mainly of vegetal remains and diatoms with low importance of animal items [40,43]. *Trichodactylus borellianus* belongs to the neotropical family of freshwater crabs, Trichodactylidae, with broad distribution in South America (from 0˚ to 35˚ S) [76]. This crab has a close relationship with the floating aquatic vegetation and exhibits little mobility [34,77]. Its natural diet is characterized by vegetal remains, algae and animal items such as oligochaetes and insect larvae [44,78]. Both prawns and crabs form abundant populations associated with the aquatic macrophytes of the floodplain littoral zone [32,77,79].

### Decapods sampling and laboratory maintenance

Specimens of decapods of varied sizes were manually collected from the environment with the aid of a hand net (500 μm mesh size) during the austral late spring (November and December, 2018). *Macrobrachium borellii* and *T. borellianus* were captured from the "Ubajay" stream (31˚33'43.45"S, 60˚30'58.73"W), Santa Fe (Argentina), from the aquatic vegetation at the shoreline of water bodies. *Aegla uruguayana* were captured from "El Espinillo" stream (31˚47'09.16"S, 60˚18'57.46"W), Entre Ríos (Argentina), through the removal of stones at the bottom of the stream and placing the hand net against the current. Decapods were transferred to the laboratory in plastic containers, where they were acclimated gradually (for at least two weeks) to the experimental conditions (temperature—24 ± 1˚C; natural photoperiod–dawn and dusk around 05:00 and 20:00, respectively; conductivity– 250 ± 10 μS/cm) in aquaria with dechlorinated and aerated tap water, with rocks and PVC tubes as shelters. During this period, decapods were fed *ad libitum* with the same fish feed used in the experiments. Two commercial feeds (Garay SRL) were used and offered separately. They were elaborated for the nutrition of omnivorous (OF) (e.g. pacú - *Piaractus mesopotamicus*) and detritivorous fish (DF) (e.g. sábalo—*Prochilodus lineatus*) (Table 1). Each feed was ground separately in a mortar and passed through sieves of 1000-μm-diameter mesh to attain a size that facilitated the ingestion by decapods. Then, feeds were stored in glass bottles at 5˚C. Every 48 hours, feces and food remains were removed from the containers by siphoning and the discharged water was supplemented. Every day pH, conductivity, dissolved solids and temperature were measured with a waterproof tester (Hanna HI98129, Romania), and dissolved oxygen was analysed with an oximeter to verify water quality (YSI Proodo SKU626281, USA).

### Experimental design and procedures

The feeding experiment was carried out using specimens of each species with variable body mass and control for each treatment. Only *A. uruguayana* had a reduced number of specimens in the experiment due to the limited wild species caught.

In total, 66 individuals were used (each type of feed was offered to 12 *M. borellii*, 9 *A. uruguayana* and 12 *T. borellianus*) plus six controls without specimens. After the acclimation

**Table 1. Proximal and elemental (C, N, P) composition of detritivorous (DF) and omnivorous (OF) fish feeds used in the experiments.**

| Ingredients | OF | DF |
|---|---|---|
| Proximal composition (g/100 g dry basis)[1] | | |
| Protein | 27 | 30.68 |
| Lipids | 3.87 | 5.00 |
| Humidity | — | 3.82 |
| Crude Fiber | 3.53 | 12.24 |
| Ash | 4.28 | 8.37 |
| P total | 0.80 | 1.24 |
| Elemental composition (g/100 g dry basis) | | |
| C | 42.4±0.3 | 41.4±0.1 |
| N | 3.8±0.1 | 4.9±0.2 |
| P | 1.1±0.2 | 1.6±0.2 |
| Elemental composition (molar) | | |
| C:N | 13.13±0.15 | 9.96±0.36 |
| C:P | 86.45±7.72 | 58.30±5.63 |
| N:P | 7.68±1.74 | 6.83±3.96 |

[1] Garay SRL (Recreo, Argentina)–nutritional values analysed by the manufacturer.—not informed by the manufacturer.

period, organisms were transferred to individual plastic recipients of 1 liter, arranged randomly, where they were left for 24 hours without food. Each recipient was provided with shelter (a small rock previously washed and chlorinated), artificial aeration and covered with a shade net to reduce stress and prevent escapes. After the fast period, fish feed was offered *ad libitum* to the corresponding treatment and left for 90 minutes. Then, each organism was removed from the plastic recipients, washed with distilled water, and transferred to glass bottles (previously cleaned with bleach to reduce the bacterial load) with 150 ml of filtered (MG-F 0.7 μm, Munktell Filter-Sweden) and dechlorinated tap water. Then, 15 ml of water samples were taken from each recipient at 30 and 60 minutes using a micropipette (5000 μl), and they were preserved at -20˚C until the analytical determination of excreted nutrients was performed. The maximum incubation time of 60 minutes was previously considered adequate because excretion rates rapidly decrease after organisms stop the feed ingestion [80]. However, water samples were taken in each recipient at 30 and 60 minutes to verify if these incubation time lapses presented a variation in nutrient excretion rates and associated ratios.

At the end of the experiment, decapods were kept without feed for 24 hours to eliminate the gut content. Then, animals were anesthetized in cold water and frozen. Subsequently, individuals were oven dried at 50˚C to constant weight, and weighted (± 1 μg) to determine dry body mass. The dried body of each individual was pulverized and homogenized in a mortar, and elementally analysed (C, N and P). This study adheres to the ethical standards [81], and ethical and legal approvals were obtained prior to the start of the study by the committee of ethics and safety in experimental work of CONICET (CCT, Santa Fe).

## Analytical and elemental analysis

Water samples from each treatment were analysed for inorganic forms of N and P, ammonium ($NH_4$-N) and orthophosphate (P-$PO_4$), respectively. $NH_4$-N was quantified through the indophenol blue method [82], and P-$PO_4$ through the ascorbic acid method [83]. To determine the

stoichiometric proportion of carbon (C) and nitrogen (N) of decapods and feeds, two subsamples of each sample were elementally analysed in a CHN628 Series Elemental Determinators (LECO ®). For total phosphorus (P) analysis, dry decapods' bodies or ground powder feed were weighted (± 1 μg) and combusted in a muffle furnace at 550˚C for a minimum of 2 hours. Then, the mass of ash was weighted and acid-digested with 25 ml of HCl 1N during 15–20 minutes in a heating plate. The digested solution was brought to 100 ml with distilled water [84] and analysed using the ascorbic-acid method [83]. C, N and P of feed and decapods' bodies were expressed as g/100g at a dry weight basis (dw) and ratios C:N, C:P and N:P were calculated using molar values.

The amount of ammonium and orthophosphate obtained in each excretion chamber was corrected by subtracting the average of nutrient concentration obtained in the control replicates (chambers without decapods) from the value obtained in decapods' excretion chambers after 30 and 60 minutes. Then, results were divided by the body mass (mg of dry weight) and by the time of incubation (0.5 or 1 hour). Mass-specific $NH_4$-N, P-$PO_4$ and $NH_4$-N:P-$PO_4$ excretion rates were expressed throughout the text as N, P and N:P excretion or mineralisation.

## Data analysis

**Within taxa analysis.** *Nutrient excretion rates*. As preliminary analysis, comparisons within taxa of nutrient excretion rates (N, P and N:P) between minutes (30 and 60) were tested with repeated measures ANOVA to select the data that was used in subsequent analysis.

Linear models were run for each species to analyze the effect of body mass, body elemental content (N, P and N:P), and type of feed on the variation of N, P and N:P excretion. The factor variable (type of feed) was included in the models as an indicator variable. Number 1 referred to DF and number 0 to OF. Before run each linear model, collinearity among body mass and each body elemental content (N, P and N:P) was calculated in each species by Pearson or Spearman correlation coefficients (depending on the assumptions of normality and homoscedasticity of the variables) with p-value test. If two variables exhibited collinearity (r > 0.70), one of both was selected as independent variable for posterior analysis. After testing for collinearities, the best linear models for each species that explained excretion variations were selected based on Akaike information criterion (AIC), starting with a full model that included type of feed, body mass, body elemental content and possible 2-terms interactions. The models were selected using the function stepAIC of the package MASS [85] with R software version 3.6.3 [86], which fit all possible subsets of the full model to find the one with the lowest AIC (S1 Table).

Normality and homoscedasticity assumptions were tested on the models and the variables used were $log_{10}$-transformed to meet assumptions of normality and heterogeneity of variances. Analyses of nutrient excretion rates within taxa were conducted with R software version 3.6.3 [86].

*Body elemental content*. Linear regressions or collinearities were analysed between body N, P, C, N:P, C:N and C:P content by relating these parameters as a function of species body mass. Variables used were $log_{10}$-transformed to meet assumptions of normality and heterogeneity of variances. The assumptions of normality and homoscedasticity on each regression were tested. If the assumptions were still not met, collinearities were run without $log_{10}$ transformed. Analyses of body elemental content within taxa were conducted with R software version 3.6.3 [86]

## Among taxa analysis

**Nutrient excretion rates.** To compare N, P and N:P excretion among taxa, ANCOVA were tested using body mass as covariate. Normality and homoscedasticity assumptions were

tested on the models. If these assumptions were not met, Wilcoxon pairwise test was run. If the assumptions of normality and homoscedasticity were met, but ANCOVA assumptions were not met (different slopes and covariate), one-way ANOVA with Tukey pairwise was run.

**Body elemental content.** Body elemental content (N, P, C, N:P, C:N and C:P) was tested among taxa with ANCOVA if the assumptions of normality and homoscedasticity of the model and if the assumptions of the equal slopes and covariate among species were met. If the assumptions of normality and homoscedasticity were not met, Wilcoxon with pairwise test was run. If the assumptions of normality and homoscedasticity were met, but ANCOVA assumptions were not met (different slopes and covariate), one-way ANOVA with Tukey pairwise test was run.

Analyses of nutrient excretion rates and body elemental content within and among taxa were conducted with R software version 3.6.3 [86]. The variables used were $log_{10}$-transformed to meet assumptions of normality and heterogeneity of variances. If the assumptions of normality and homoscedasticity were still not met after transforming, data without $log_{10}$-transformed were used in the non-parametric stats.

Finally, the relationship between $log_{10}$-transformed variables (body mass and body elemental content) and the nutrient excretion was analysed through Principal Component Analysis (PCA), based on a correlation matrix, as a conclusion and visualization of the main results. This analysis included nutrient excretion rates and ratios of the selected time (30 or 60 minutes), body elemental contents (single elements and ratios), and body mass, considering the taxonomic identity of each species.

## Results

The elemental composition of DF exhibited a slightly higher amount of N (4.9 ± 0.2%) and P (1.6 ± 0.2%) compared to the OF (3.8 ± 0.1% and 1.1 ± 0.2%, respectively). The percentages of C of both feeds were comparatively similar while the contents of C:N and C:P of OF were slightly higher than DF (Table 1).

Mean body masses (dw) of decapods used in the experiments were *M. borellii* (351.0 ± 82.1 mg ind$^{-1}$), *A. uruguayana* (554.2 ± 536.8 mg ind$^{-1}$) and *T. borellianus* (110.8 ± 50.7 mg ind$^{-1}$), with ranges of [44.3–377.6 mg ind$^{-1}$], [8.2–1900.1 mg ind$^{-1}$] and [26.7–220.9 mg ind$^{-1}$], respectively.

### Within taxa

**Nutrient excretion.** Individuals of *M. borellii* did not show significant differences in the N, P and N:P excreted at 30 and 60 minutes (F: 1.9800, $p$ = 0.1730; F: 2.9420, $p$ = 0.1020, F: 2.2290, $p$ = 0.1500 respectively). *Aegla uruguayana* excreted N faster over 30 minutes (0.95 ± 1.63 μg N mg dw$^{-1}$.hr$^{-1}$) than over 60 minutes (0.31 ± 0.35 μg N mg d w$^{-1}$.hr$^{-1}$) and these differences were statistically significant (F: 18.789, $p$ = 0.0005). This species presented similar P (F: 3.7510, $p$ = 0.1480) and N:P (F: 0.0320, $p$ = 0.8640) excretion in both incubation times. On the other hand, in *A. uruguayana* many data were registered in which the presence of P at 60 minutes was very close to the analytical detection in contrast with the data registered at 30 minutes. Individuals of *T. borellianus* did not show significant differences with N (F: 0.1210, $p$ = 0.7320, respectively), P (F: 0.2230, $p$ = 0.6430) and N:P excreted in both incubation times (F: 0.1480, $p$ = 0.7060 for *T. borellianus*). Considering these preliminary results, all posterior analyses were made using the data of nutrient excretion rates obtained at 30 minutes.

In *M. borellii* there were no collinearities among body mass and body N, P and N:P content (r < 0.70; p > 0.05). The variables that most affected N excretion of prawns were body mass and body N content, both with negative relationship (Table 2) (Fig 1A and 1D), while no variable explained P and N:P excretion variations in this species (Table 2).

**Table 2. Results of linear models within species (with log$_{10}$-transformed data) chosen by Akaike Information Criterion (AIC) to explain the nutrients excretion rates and ratio.**

| | AIC | Coefficient | T-value | p-value |
|---|---|---|---|---|
| *M. borellii* | | | | |
| • Excretion N | | | | |
| **~ Body mass + N body** | -40.55 | | | |
| Body mass | | -0.6852 | -3.4340 | 0.0074* |
| N body | | -8.1662 | -3.4350 | 0.0074* |
| • Excretion P | | | | |
| **~ Feed + Body mass + P body + Feed\*P body** | -48.26 | | | |
| DF | | -0.4877 | -1.4150 | 0.1774 |
| Body mass | | -0.3200 | -1.3900 | 0.1849 |
| P body | | -2.0918 | -1.0540 | 0.3085 |
| DF\*P body | | 5.4611 | 2.0550 | 0.0577 |
| • Excretion N:P | | | | |
| **~ Feed\*Body mass + Feed\*N:P body + N:P body\*Body mass** | -29.24 | | | |
| DF | | -4.5730 | -0.8530 | 0.4840 |
| Body mass | | 21.2440 | 1.1070 | 0.3830 |
| N:P body | | 34.4140 | 0.9670 | 0.4350 |
| DF\*Body mass | | -1.4280 | -0.8780 | 0.4720 |
| DF\*N:P body | | 6.4200 | 2.8430 | 0.1050 |
| N:P body\*Body mass | | -16.6730 | -1.0990 | 0.3860 |
| *A. uruguayana* | | | | |
| • Excretion N | | | | |
| **~ Feed + Body mass + N body + Feed\*Body mass + Feed\*N body** | -38.23 | | | |
| DF | | -7.9841 | -1.3940 | 0.2127 |
| Body mass | | -0.6875 | -5.3650 | 0.0017 ** |
| N body | | -3.5236 | -1.7840 | 0.1246 |
| DF\*Body mass | | -0.5446 | -1.3570 | 0.2236 |
| DF\*N body | | 11.8409 | 1.8900 | 0.1076 |
| • Excretion P | | | | |
| **~ Feed + Body mass + P body + Feed\*Body mass + P body\*Body mass** | -37.69 | | | |
| DF | | -2.1696 | -2.9700 | 0.0178 * |
| Body mass | | -1.5187 | -7.2910 | 8.5x10$^{-05}$ *** |
| P body | | -24.9488 | -1.8880 | 0.0956 |
| DF\*Body mass | | 1.0006 | 3.6220 | 0.0067 ** |
| P body\*Body mass | | 9.4858 | 2.0500 | 0.0745 |
| • Excretion N:P | | | | |
| **~ Feed + Body mass + N:P body + Feed\*N:P body + N:P body\*Body mass** | -24.90 | | | |
| DF | | -19.879 | -2.5010 | 0.0667 |
| Body mass | | 9.7620 | 1.4980 | 0.2084 |
| N:P body | | 10.8110 | 0.8760 | 0.4305 |
| DF\*N:P body | | 18.3200 | 2.4800 | 0.0682 |
| N:P body\*Body mass | | -9.2240 | -1.5310 | 0.2006 |
| *T. borellianus* | | | | |
| • Excretion N | | | | |
| **~ Feed + Body mass + N body + Feed\*Body mass + N body\*Body mass** | -39.45 | | | |

*(Continued)*

**Table 2.** (Continued)

| | AIC | Coefficient | T-value | p-value |
|---|---|---|---|---|
| DF | | -2.9161 | -3.4550 | 0.0259* |
| Body mass | | -8.5637 | -0.9400 | 0.4006 |
| N body | | -20.9838 | -0.8740 | 0.4316 |
| DF*Body mass | | -1.5021 | -3.5790 | 0.0232* |
| N body*Body mass | | 11.9194 | 1.0390 | 0.3576 |
| • Excretion P | | | | |
| **~ Feed + Body mass + P body + Feed*P body + P body*Body mass** | 0.36 | | | |
| DF | | 5.0550 | 1.2350 | 0.3050 |
| Body mass | | -28.6050 | -2.2030 | 0.1150 |
| P body | | -244.320 | -2.0280 | 0.1360 |
| DF*P body | | -22.0670 | -1.3040 | 0.2640 |
| P body*Body mass | | 112.0250 | 2.0060 | 0.1380 |
| • Excretion N:P | - | - | - | - |

Statistically significant differences

*$p < 0.05$

**$p < 0.005$

***$p < 0.001$.

DF: Detritivorous feed.

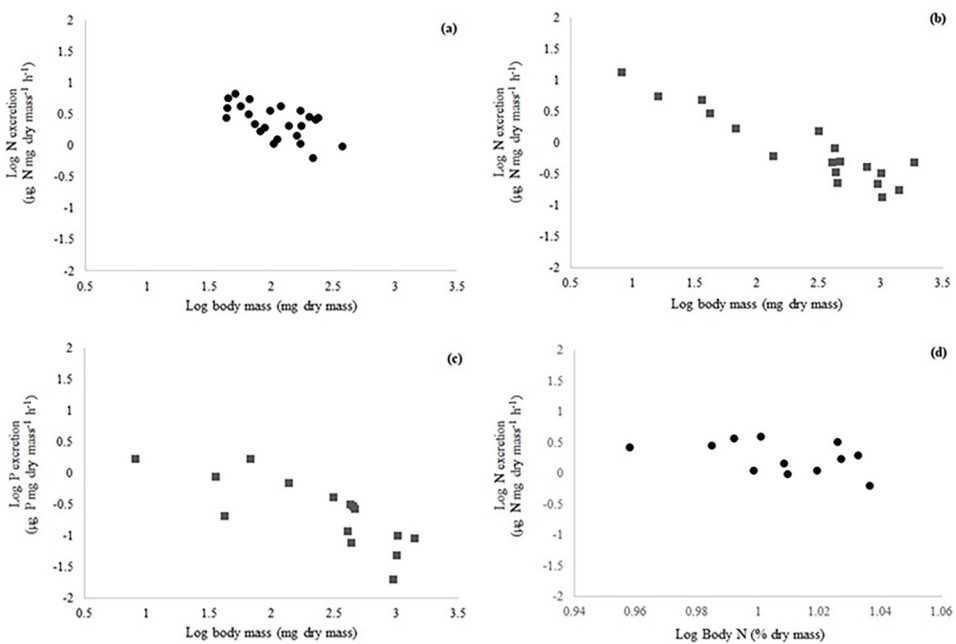

**Fig 1.** Within taxa relationships between N excretion and body mass of *Macrobrachium borellii* (a) and *Aegla uruguayana* (b), between P excretion and body mass of *Aegla uruguayana* (c); between N excretion and body N content of *Macrobrachium borellii* (d). *Macrobrachium borellii* (black circle) (a, d), *Aegla uruguayana* (dark gray square) (b, c). The excretion rates values were expressed by hour, but the measures were taken at 30 minutes.

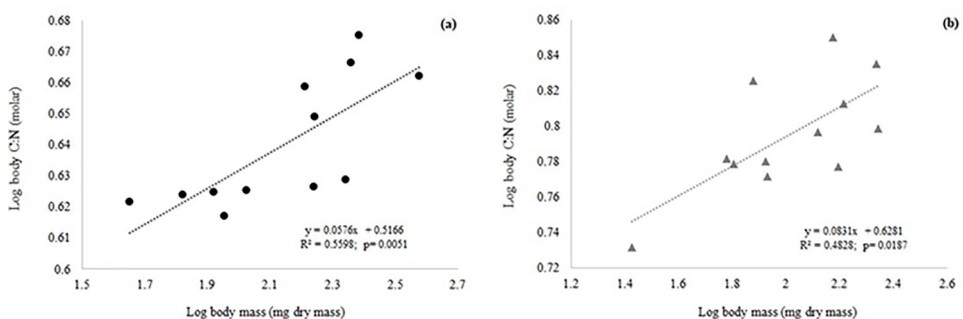

**Fig 2.** Linear regressions within taxa of body content of C:N as a function of invertebrate body mass of *Macrobrachium borellii* (a), *Trichodactylus borellianus* (b). *Macrobrachium borellii* (black circle) (a), *Trichodactylus borellianus* (light gray triangle) (b).

In *A. uruguayana*, there was no collinearity between body mass and body elemental contents ($r < 0.70$, $p > 0.05$). Body mass influenced the N excretion variations in this species with negative relation (Table 2) (Fig 1B). The P excretion was affected by the type of feed, body mass and the interaction between these variables (Table 2), decreasing with increasing body mass (Table 2) (Fig 1C). This species excreted more P when were fed with OF (0.4943 ± 0.6129 µg P mg dry mass$^{-1}$ h$^{-1}$) than with DF (0.4120 ± 0.5100 µg P mg dry mass$^{-1}$ h$^{-1}$). According to body mass (interaction term), the slope of P was higher with OF (1.2152), than with DF (0.4201), both with negative slope. No variable explained N:P excretion variations in this species (Table 2).

In *T. borellianus* there was no collinearitiy between body mass and body elemental contents ($r < 0.70$; $p > 0.05$). Type of feed and the interaction of type of feed with body mass influenced the N excretion variations (Table 2). The N excretion was higher when crabs were fed with OF than with DF (0.6189 ± 0.3555 and 0.5865 ± 0.4798 µg P mg dry mass-1 h-1, respectively). According to the significance between interaction term, the relation between N excretion and body mass when crabs were fed with OF was positive (slope: 0.5648) and with DF, negative (slope: -0.8569). No variable explained P excretion variations in this species (Table 2). Regarding the model of N:P excretion, there were many NA data of body N:P content of *T. borellianus*. When the full model was run, there were NA statistical relations between dependent and independent variables and between independent variables that interacted. Therefore, the full model failed to run with AIC (Table 2). For this reason, the full model was separated in three models of N:P excretion vs. each one interaction with predictor variables (Feed*Body mass; N:P body*Body mass; Feed*N:P body). However, no effects were identified in N:P excretion model when were run separately ($p > 0.05$).

**Body elemental content.** *Macrobrachium borellii* and *T. borellianus* showed a significant and positive linear relationship between body C:N content and body mass (Fig 2A and 2B). Also, only *T. borellianus* showed a significant negative allometric relationship between body N content and body mass, but at the limit of the significance ($p = 0.0483$; $R^2 = 0.3668$; $b = -0.0927$). Variations in body P, and C, N:P and C:P content did not show significant linear relationships ($p > 0.05$) and neither collinearities ($r < 0.70$; $p > 0.05$) with body mass in any species of decapods.

## Among taxa analysis

**Nutrient excretion rates.** Despite N and P excretion showed equal slopes among taxa (ANCOVA $p > 0.05$), decapod species had different body mass. This means that the covariate

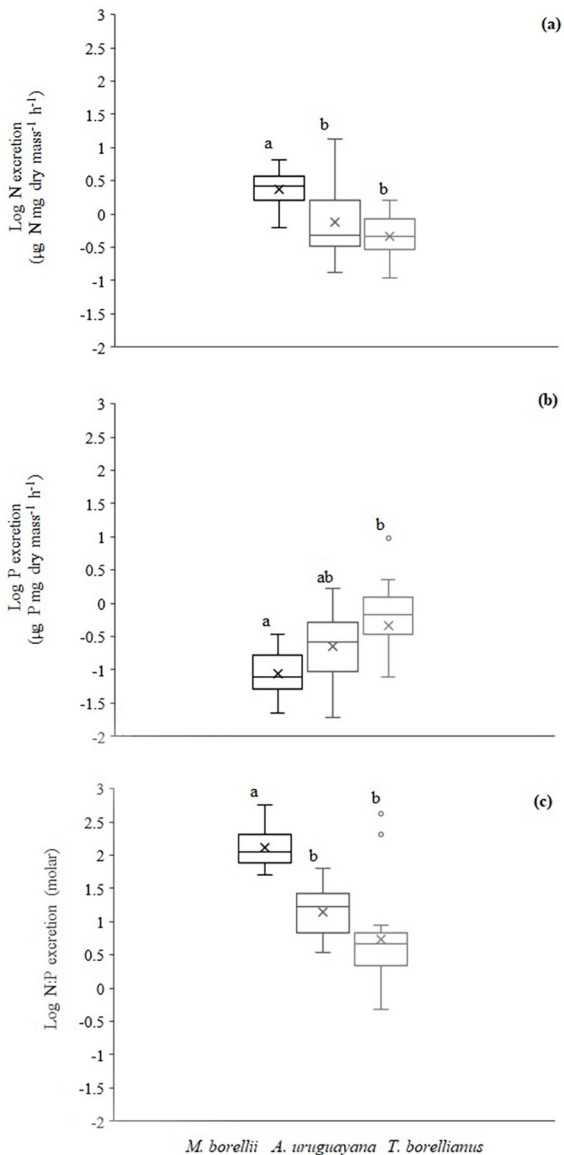

**Fig 3.** Box plots of excretion rates of N, P and N:P of each decapod species (a, b, c). The top, bottom, and line through the middle of the boxes correspond to the 75th, 25th, and 50th (median) percentile. The whiskers extended from the 10th percentile to the 90th percentile. Crosses indicate the median values. Different letters above bars indicate significant differences among taxa ($p < 0.05$). The excretion rates values were expressed by hour, but the measures were taken at 30 minutes.

among taxa was different and the assumptions of ANCOVA were not met. According to ANOVA test, the N excretion was different between *M. borellii* and the other species. N excretion of *M. borellii* was higher in comparison to *A. uruguayana* (Tukey *post hoc*, $p = 0.0002$) and *T. borellianus* (Tukey *post hoc*, $p < 0.001$) (Fig 3A). *Macrobrachium borellii* showed highest values, *A. uruguayana* intermediate, and *T. borellianus* lowest values of N excretion (Fig 3A). The P excretion followed the opposite trend with *M. borellii* with lowest values and differed significantly with *T. borellianus* (Tukey *post hoc*, $p = 0.0164$) (Fig 3B). The N:P excretion was higher in *M. borellii* than in *A. uruguayana* (Wilcoxon pairwise, $p = 0.004$) and *T. borellianus* (Wilcoxon pairwise, $p = 0.0072$) (Fig 3C).

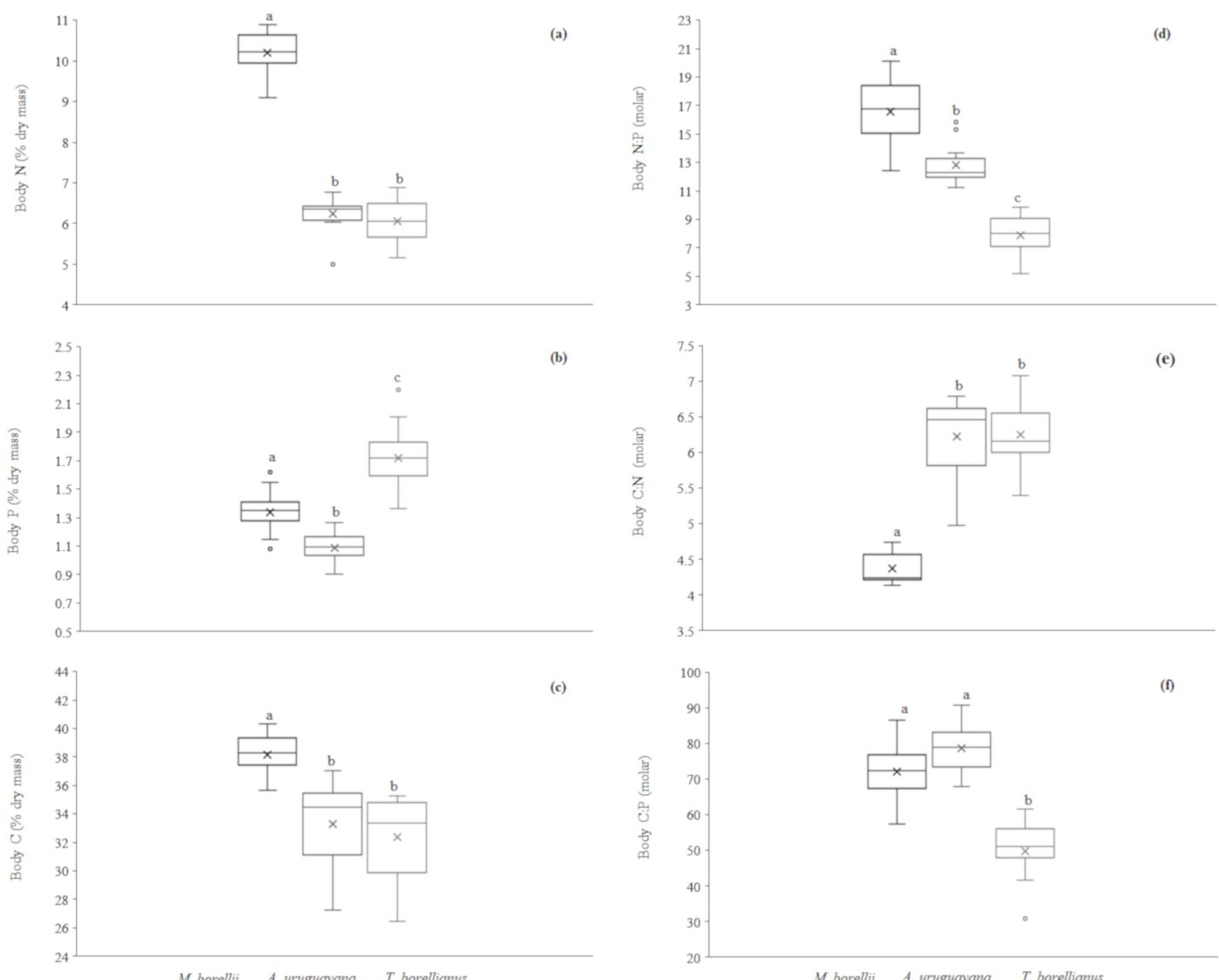

**Fig 4. Box plots of body content of N, P, C and N:P, C:N, C:P of each decapod species.** The top, bottom, and line through the middle of the boxes correspond to the 75th, 25th, and 50th (median) percentile, respectively. The whiskers extended from the 10th percentile to the 90th percentile. Crosses indicate the median values. Different letters above bars indicate significant differences among taxa ($p < 0.05$).

**Body elemental content.** Similarly to the previous analysis, the assumptions of ANCOVA were not met because decapod species had different body mass (covariate). According to the *post hoc* analysis, body N, C and C:N contents were statistically different between *M. borellii* and the other species. Prawns exhibited highest body N content with respect to the other decapods (Tukey *post hoc*, $p < 0.001$, in both cases) (Fig 4A) and the highest body C content with respect to *A. uruguayana* (Wilcoxon pairwise, $p = 0.0001$) and *T. borellianus* (Wilcoxon pairwise, $p = 4.4000 \times 10^{-06}$) (Fig 4C). The body C:N content of *M. borellii* was significantly lower in comparison to *A. uruguayana* and *T. borellianus* (Tukey *post hoc*, $p < 0.001$, in both cases) (Fig 4E). The percentage of body P and N:P contents were statistically different among all species (Tukey *post hoc*, $p < 0.001$ in all cases). The crab *T. borellianus* showed the highest body P content and *A. uruguayana*, the lowest one (Fig 4B). Moreover, *M. borellii* showed the highest

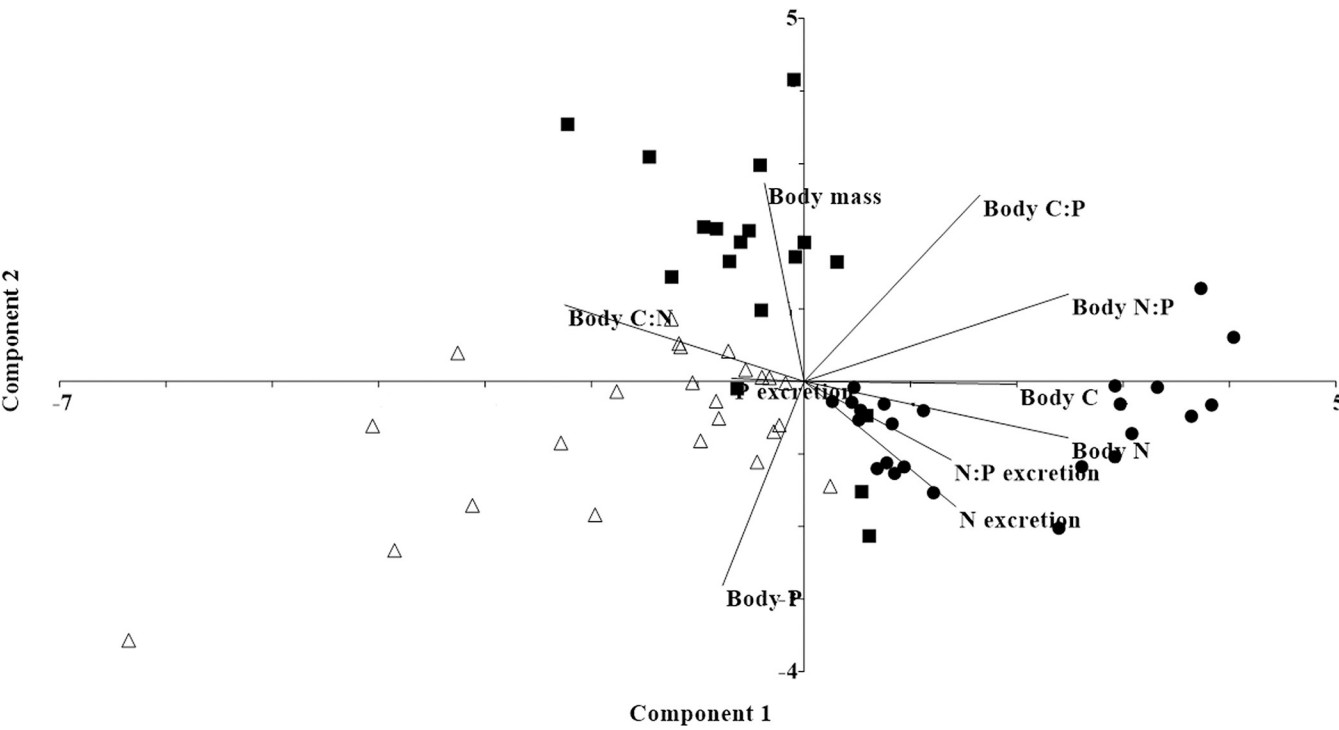

**Fig 5. Principal component analysis with log10-transformed of nutrient excretion rates (N and P) and ratio (N:P), body elemental contents (body N, P, C and body N: P, C:N, C:P) and body mass.** *Macrobrachium borellii* (black circle), *Aegla uruguayana* (dark gray square), *Trichodactylus borellianus* (open triangle).

body N:P content and *T. borellianus*, the lowest one (Fig 4D). Finally, *T. borellianus* showed the lowest body C:P content in comparison to the other species (Wilcoxon pairwise, $p < 0.001$) (Fig 4F).

**PCA overview.** The first two components of PCA explained the 38.66% (Component 1) and 22.29% (Component 2), showing variations among species (Fig 5). Body N, C, N:P and C: N contents characterised species' variations with high correlation values in Component 1 (Table 3). Body N, C, and N:P contents were higher in *M. borellii* and lower in the other crustaceans, while body C:N followed the opposite trend (Fig 5). Body P content showed high correlation values in Component 2 (Table 3), being higher in *T. borellianus* and lower in *A. uruguayana* (Fig 5). The body C:P content indicated similar values of correlation in both components (Table 3), and was low in *T. borellianus* (Fig 5). Nutrient excretions and ratio presented low values of correlation in both components (Table 3). However, it was observed that N excretion increased in *M. borellii*. Besides, this variable increased with body N content in this species (Fig 5). It was observed that N excretion decreased with body mass. *T. borellianus* exhibited higher P excretion and the N:P excretion increased in *M. borellii*. Body mass presented the higher correlation in Component 2, and was higher in *A. uruguayana* (Table 3) (Fig 5). This anomuran showed an intermediate separation of the other species in the multivariate space (Fig 5).

## Discussion

In this study, we found that body mass and body content mainly influence the role of crustacean decapods as nutrient recyclers. While body mass was the main factor that explained the variation in nutrient excretion within taxa, body content and body mass explained the

**Table 3. Correlation values of excretion rates (N, P) and ratio (N:P), body nutrient contents (N, P, C) and ratios (N:P, C:N, C:P), and body mass on the first and second principal components of PCA-Principal component analysis.**

| Variable | Correlations of PCA | |
|---|---|---|
| | **Component 1** | **Component 2** |
| N excretion | 0.5185 | -0.4657 |
| P excretion | -0.2482 | 0.0111 |
| N:P excretion | -0.5026 | -0.2916 |
| Body N content | 0.9026 | -0.2097 |
| Body P content | -0.2788 | -0.7587 |
| Body C content | 0.7498 | -0.02835 |
| Body N:P | 0.9034 | 0.3247 |
| Body C:N | -0.8188 | 0.2841 |
| Body C:P | 0.6003 | 0.6918 |
| Body mass | -0.1354 | 0.7369 |

differences found among taxa. The hypothesis and predictions stated previously were partly met as detailed below. Within taxa, **1**- body mass, body elemental content and feed predicted nutrient excretion rates and ratio with different predictive power. **1a**- Body mass best explained the variation in nutrient excretion but only in prawns (N excretion) and anomurans (N and P excretion), **1b**- scaled negatively, and **1c**- there was no relation between N:P excretion rate and N:P body content. These results provide data that partially supports the MTE and the EST theories but not the Sterner's postulation [12]. **2**- Body C:N content positively scaled with body mass in prawns and crabs, while body N scaled negatively only in crabs. **2a**- Body P content had no relation with body mass. These results did not support the GRH.

Considering the separation of species in the multivariate space, we observed that body content followed by body mass were the main factors that explained the differences among taxa, while the excretion rates had lower predictive power. **3**- N, P and N:P excretion were quite different. Prawns mineralised more N and less P and, consequently more N:P, than anomurans and crabs. Body elemental content were also quite different among taxa. Prawns exhibited highest body N and C content and lower body C:N, while crabs showed the highest body P and lowest C:P. **3a**- Body C:N was higher in most carcinized species (*A. uruguayana* and *T. borellianus*) while body N:P is higher in prawns, the species with more carnivorous trophic habit.

We discussed the implications of the main results, their relations with ecological theories and the usefulness in productive aquatic systems such as IMTA.

## Within taxa

**Nutrient excretion rates.** An important preliminary concern to discuss are the experimental conditions. We noticed a decrease in N excretion rates in *A. uruguayana* with time. Previous studies had also reported the reduction in mineralisation rates with time and attributed to fasting and handling stress [15,21,22,80]. In our research, decapods were acclimated to experimental conditions for 2 weeks and fed the same feed during this period and just before the excretion trial. In this way, we could infer that differences found in N excretion time had low bias due to fasting and handling stress. Because the time of incubation affected the N excretion of anomurans, and there was a general trend in the other decapods to excrete more during the first 30 minutes of incubation, we concluded that this period is most adequate than 60 minutes to make comparative analyses with the studied decapods. In addition, because we

did not measured nitrates, 30 minutes of incubation plus standardization of methodology could reduce the error of estimating N excretion due to the action of nitrifying bacteria that oxidize ammonia to nitrites and nitrates [15]. In addition, the fact that more variables in all species explained the N excretion in comparison with the P excretion could be due to the animals' necessity to rapidly remove toxic ammonia from protein metabolism, which negatively influences the organisms' homeostasis [87].

Prawns and anomurans presented a negative and significant relation between N excretion and body mass. These findings are in accordance with MTE. The non-significance of this relationship in crabs may be due to the narrow size range of the specimens used in the present study in comparison with the other species. Several studies have shown that body size (by extension, body mass) is the best predictor of N and P excretion rates in fish, decapod crustaceans and other invertebrates [18,20,22,88–90]. This relationship is even more marked in the excretion of N than in P [6], such as observed in the present study.

The importance of body elemental composition and diet in predicting excretion rates are key concepts in EST. Our results indicated that these variables had a limited predictive power for our data, such as also observed by [15,20]. Only the body N content of *M. borellii* showed a positive relation with N excretion rate, which are in contradiction with the negative relation theoretically expected [5]. Furthermore, we did not find significant relation between body N:P and excreted N:P to corroborate the Sterner's postulation [12]. The greater predictive power of body mass in comparison to body elemental content was in concordance with previous studies [6,20] that found little support for the predictions of EST in aquatic organisms. This is reasonable since the excretion of waste nutrients is related to the acquisition and assimilation of food, both processes driven by metabolic demands that change during the life cycle of the organisms [6,7]. Although several studies highlighted the effect of body elemental content on excretion rates in various groups of animals [13,88], for the analysed decapod species, this trait has a weak predictive value [6,18,20].

Feed predicted the nutrient excretion only in few cases (two of six) and showed interaction with the biomass. This result are in accordance with other studies that found diet as a weak predictor of excretion rates [15,17]. Regardless of the biomass, anomurans excreted more P and crabs excreted more N when fed OF. When body mass interacted with excretion, anomurans had negative allometry for P excretion and crabs positive allometry for N excretion. In theory, any element in excess of the appropriate ratio should be discarded [8]. Feed C:N and C:P contents exceed in both cases these elemental ratios of anomurans and crabs bodies, respectively, while N:P is limiting in feeds. Additional studies of crustaceans' egestion are necessary to determine if the release of non-limiting nutrients could be due to an inefficient digestion and assimilation of N and P from feed [95 and cites herein]. In addition, *A. uruguayana* exhibits faster egestion rates than *M. borellii* when fed OF (Musin et al. unpublished). These differences could be related to a trade-off between faster feeding and assimilation efficiency in each species [95 and cites herein].

With respect to the observed allometry, it is expectable that juveniles need more N and P for growth [8] and, therefore, the relation between excretion rates of both elements and biomass might be positive, which only occurred in N excretion. Again, these unexpected results might be elucidated with the nutrient composition of feces as long as it is methodologically feasible to collect feces from juvenile individuals. The trial time to collect the minimum amount of feces (mg dry basis) for elemental nutrient determination is sometimes intangible even in adult individuals, such as in *M. borellii* (Musin et al. unpublished).

**Body elemental content.** Within taxa analysis revealed that body C:N content scaled positively with body mass in *M. borellii* and *T. borellianus*, while in this last species body N scaled negatively with body mass (at the edge of significance). Adult crabs had less N than juveniles.

This finding could be related to the relative importance of animal items in the natural diet of this species. Larger individuals consume more vegetal remains than smaller ones, which consume more insect larvae and oligochaetes [44,78]. This feeding habit is also in congruence with the positive allometry of body C:N observed in this crab, since adults eat more C-rich items. We also observed a higher body C:N in adults of *M. borellii*. However, prawns seem to maintain a preference for animal components during the entire life cycle, contradicting this allometric tendency of body C:N content. If there was no evidence that diet could drive variation in body stoichiometry during ontogeny, this relationship could be related to differences in the proportion of chitin between juveniles and adults. Since the chitin of the exoskeleton has high C:nutrient [8,27,91–93], adults of both crabs and prawns should have higher amounts of C than N due to more robust carapace. It is necessary to complement these results with studies that explore the influence of diet on body stoichiometry through ontogeny and the deviation from strict homeostasis to get a better understanding of nutrient allocation strategies. We also recommend a complementary study with a broad range of body mass of each species (mainly for *T. borellianus*) to confirm the allometric trends of body elemental content found in our research.

## Among taxa

**Nutrient excretion.** Although nutrient excretion rates were not the main factors separating the species in multivariate space, there were significant differences of excretion rates among taxa in univariate analysis. The nutrient excretion rates of *M. borellii* were quite different from the other decapods. Prawns mineralised more N, less P, and more N:P, contradicting what would be expected from the EST predictions. Even with higher body N content and after the consumption of diets that did not fulfill the protein requirement of this species to grow (~35%) [48], prawns mineralized higher N than the other species. Anomurans and crabs released the intermediate and the lowest rates of N (without significant differences between them), respectively, showing the same trend in their respective amounts of body N. These results did not align with EST predictions [8] either and were surprising. As suggested by [94] in armour fish taxa, these results could be related to the variation in the assimilation efficiency, and even in variations in the phenotype expression.

Future studies should include egestion rates of these organisms and feed digestibility to help understand differences in assimilation efficiency in each taxon [95]. These feeding trials would also be helpful to quantify elemental imbalances between body and diet, and to verify the homeostasis regulation of these species [8]. Such trials should consider the residence time and turnover of elements in decapods' tissues [96] in the experimental design. Regarding P excretion rates, *M. borellii* excreted the lowest amount of this nutrient. In heterotrophic organisms, the assimilated P is stored as polyphosphate rich in energy [97,98]. Therefore, prawns would assimilate a greater amount of this nutrient because they are more active than crabs and anomurans, and might have higher energy expenditure in mobility. Despite the complexity to understand the differences observed in excretion rates among species and the need to complement the results with additional studies, our findings emphasize the importance of the taxonomic identity in the animal-mediated nutrient cycling [10].

The comparison of nutrient excretion rates with other freshwater decapod species is a difficult task due to the limited studies in this taxonomic group, different methodological approaches (e.g. laboratory and nature) and variable units of measure. For example, the N excretion rate of *M. olfersii* seems to be quite higher than *M. borellii* (around 2.7 and 0.5 of median values, respectively) [17]. However, while the study with *M. olfersii* was made in the field and excretion rates were calculated using wet weight and expressed as µg g $WM^{-1}$ $h^{-1}$, our

results were made in laboratory using dry weight and expressed as µg mg dry mass$^{-1}$ h$^{-1}$. We observed a similar problem in another field study with two species of crustaceans (*Xiphocaris elongata* and *Atya spp*) captured with a backpack electrofisher (very stressful methodology) and which results were expressed as log mol NH$_4$-N shrimp$^{-1}$ h$^{-1}$. A study with other freshwater prawn, *M. acanthurus* [88], reported lower values of N (below 0.1 g N mg dw$^{-1}$h$^{-1}$ average values) and P (below 0.01 g N mg dw$^{-1}$h$^{-1}$of average values) excretions in comparison with *M. borellii* (around 0.35 and -1 log g N mg dry mass$^{-1}$h$^{-1}$, respectively, of average values). Although both studies were realized in laboratory and used similar units of measure, they were methodologically different once *M. acantharus* was not fed with artificial feed [88]. These data emphasize the need to use homogeneous methodologies and units to express results among studies.

**Body elemental content.** The main differences among taxa were body elemental contents evidenced in the multivariate space and univariate analysis. Prawns had higher body N and C contents, as well as lower C:N than anomurans and crabs. These findings could be related to the natural diet and the carcinization of crustaceans. *Macrobrachium borellii* have an omnivorous trophic habit with preference for animal preys [40,45]. This species requires high protein content to achieve better growth parameters [48] and it is an uncarcinized decapod. These traits could contribute to a lower body C:N content compared to the other species, due to a higher ingestion of N-rich food items (higher trophic status) and to a less chitinous carapace. Moreover, digestive enzyme studies revealed that *M. borellii* had higher proteinase activity while juveniles of *A. uruguayana* had higher amylase activity in hepatopancreas [56], indicating that metabolism of nitrogen and carbohydrates, respectively, predominates in these organisms. The high body C content of *M. borellii* could be explained by the fact that prawns store more lipids in muscle tissue than anomurans [56] and that they also have comparatively more muscle tissue (thus, more body N), which might contribute to the lower values of body C:N. On the other hand, crabs-like body forms species (robust carapace with lower N-rich bodies), such as *T. borellianus* and *A. uruguayana*, also consume more algae and vegetal remains than prawns [40,46,96]. The lower trophic status of crabs and anomurans, therefore, exhibited more C:N, as expected.

Furthermore, body P and N:P content were different among the three species, with *T. borellianus* presenting the higher and lower values, respectively. This crab seems to have higher specific growth rates than prawns and anomurans when fed the same omnivorous feed used in this study (Calvo et al., unpublished data). Therefore, higher body P content should be related with a greater allocation of this element to the production of P-rich rRNA to hold faster growth [4], and the lowest N:P ratio is explained by the high P content with respect to N in this species' body. However, this hypothesis should be corroborated by carrying out growth experiments that ensure there is no deficiency in any element regarding the feed formulation, and that it matches the nutritional requirements of tested species.

The similarity of N and P composition of *M. borellii* (10.2% of body N, 1.3% of body P, and 16.6% of body N:P) with *M. olfersii* (around 10.5% of body N, 1.5% of body P, and 17% of body N:P) [17] emphasizes the role the taxonomic identity in the separation of species. Both species belongs to the same genus and had omnivorous trophic habit, which might explain the high similarity in comparison with other freshwater prawns-like species, like *Xiphocaris elongate*,and *Atya spp* [18]. However, the body content of *M. acanthurus* seems to be quite different (5‰ of body N, 1–2‰ of body P) [88] from both *Macrobrachium spp*, but we suspect that it could be a mistake in the data report.

**Comments about application in IMTA.** Aquaculture represents almost 50% of the fishery products produced globally and continues to grow faster than other food production sectors, mainly in freshwater systems [98]. The nutrient discharge of this food production system

has generated many environmental impacts on ecosystems [99] that could be mitigated and recycled through the diversification and incorporation of complementary species, such as proposed by IMTA concepts [100,101]. In low taxa diversity aquatic systems, like an IMTA, the nutrient turnover is strongly influenced by the few taxa that compose it [14,69]. Therefore, understanding the physiological processes of organisms is essential to explore and predict nutrient recycling behavior in different scenarios and to design integrated productive systems that efficiently use the nutrient inputs through an adjusted mass balance.

Many studies have shown that the inefficiency in nutrient recovery due to feed input in aquaculture systems could be mitigated by the diversification and integration of complementary species [102–104]. However, the use of native freshwater decapods as non-fed species is still a topic with fewer studies. The integration of prawns as organic nutrient recyclers in fish culture could enhance the nutrient recovery and the harvested biomass such as observed with *M. amazonicum* in fish-prawn IMTA system with *Colossoma macropomum* [105–107] and *Astyanax lacustris* [108]. According to our results, prawns and anomurans exhibited allometry in N recovery per mass unity. Therefore, juveniles would contribute more to transforming N from proteins to ammonium (i.e. reducing organic burden) and enhance the availability of this compound in the water column for nitrifying bacteria to finally grow vegetables or algae. On the other hand, prawns incorporated more P and, therefore, could limit this nutrient to primary producers while representing a P-rich by-product for many purposes. Contrary, crabs mineralised less N and more P exerting the opposite effect.

The role of decapods as animal-mediate nutrient recyclers is different among taxa, depends mainly on body mass and should be considered to select complementary species that efficiently use feed resources. The decision to use, or not, a particular crustacean species to integrate an IMTA system could change the nutrient turnover from one compartment (pond bottom due to sedimentation of non-fed feed) to another (mineralized inorganic nutrient in the water column) and the quantity and quality of N and P available in the water. Moreover, due to benthonic habit of this species, the bioturbation action might increase the N and P uptake from microorganisms present, for example, in the biofilm [109,110].

Body elemental content is another variable that could be a useful baseline to estimate the nutritional requirements of animals [111,112]. Our study revealed that this variable differed with body mass (ontogeny) and that it is related to the natural trophic status of organisms. Results showed higher N-requirements and higher trophic status (lower C:N) of prawns than anomurans and crabs. Moreover, prawns and crabs changed the trophic status during ontogeny, requiring lower N in comparison with C at later stages. Prawns needed more protein in the feed, and might be successfully integrated with fish of higher N-requirements, while crabs and anomurans might exhibit good performance with fish that are fed lower N-rich diets, such as herbivorous and omnivorous fish.

Our study was an effort to contribute to the ecological background of MTE and EST through empirical data obtained from freshwater decapod species with aquacultural potential use, and to provide useful information about nutrient mineralisation and nutritional requirements based on body elemental content of these decapods. These results, added to other empirical data on egestion, digestibility, retention in biomass (growth and reproduction), and food intake, offer a framework to leave some open questions for further studies and provide information to infer the amount of decapods needed to biomitigate the feed remains of detritivorous and omnivorous fish culture, through an improved mass balance. In addition, contributes to the background information about no traditional, native and no fed species with potential in IMTA. Ecological theories and experimental aquaculture research can be good allies when it is necessary to "imitate" nature and achieve more efficient production processes with less impact on the ecosystem.

## Supporting information

**S1 Table. Results of Akaike Information Criterion (AIC) of all model subsets (with log$_{10}$-transformed data) using stepAIC function.**
(DOCX)

## Acknowledgments

We thank C. De Bonis for his field assistance, M. C. Mora for her lab assistance, F. Giri for his statistical assistance, and the academic editor Dr. David B. Lewis for his valuable contribution in revising this manuscript.

## Author Contributions

**Conceptualization:** Débora de Azevedo Carvalho.

**Data curation:** Débora de Azevedo Carvalho.

**Formal analysis:** Gabriela Musin, María Victoria Torres, Débora de Azevedo Carvalho.

**Funding acquisition:** María Victoria Torres, Débora de Azevedo Carvalho.

**Investigation:** Gabriela Musin, María Victoria Torres, Débora de Azevedo Carvalho.

**Methodology:** Gabriela Musin, María Victoria Torres, Débora de Azevedo Carvalho.

**Project administration:** María Victoria Torres, Débora de Azevedo Carvalho.

**Resources:** Débora de Azevedo Carvalho.

**Supervision:** Débora de Azevedo Carvalho.

**Validation:** Gabriela Musin, María Victoria Torres, Débora de Azevedo Carvalho.

**Visualization:** Gabriela Musin, María Victoria Torres, Débora de Azevedo Carvalho.

**Writing – original draft:** Gabriela Musin, María Victoria Torres, Débora de Azevedo Carvalho.

**Writing – review & editing:** Gabriela Musin, María Victoria Torres, Débora de Azevedo Carvalho.

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
