## [Decision Letter · Decision Letter 0]

13 Jun 2022

PONE-D-22-00145Consumer-driven nutrient recycling of freshwater decapods: linking ecological theories and application in integrated multi-trophic aquaculturePLOS ONE

Dear Dr. Carvalho,

Thank you for submitting your manuscript to PLOS ONE. After careful consideration, we feel that it has merit but does not fully meet PLOS ONE’s publication criteria as it currently stands. Therefore, we invite you to submit a revised version of the manuscript that addresses the points raised during the review process. In addition to (and in agreement with) the reviewer comments, I provide several of my own observations here. The questions and hypotheses were a bit confusingly laid out. In particular, questions 1 and 2 seem redundant. Both questions appear to ask how N, P, and N:P excretion are influenced by a list of several predictor variables. I think you may have distinguished these as two questions to match the two different statistical approaches (mixed-effects models vs. ANCOVA regressions). But ultimately, they seem like the same question to me: how do a bunch of variables (body size, etc.) affect excretion chemistry. Perhaps you can simplify your question statements, followed by a list of predicted relationships (what you currently refer to as hypotheses). I agree with both authors that, overall, the manuscript was too long. The Discussion alone is nearly 3500 words. Perhaps you can focus the Discussion text on a few key findings. I share reviewer 1’s suspicion that you have included too much. Perhaps consider just presenting the species-specific analyses. There are strong species differences in excretion and body nutrient content and ratios, and the species are different sizes (and size also predicts excretion and body nutrient content and ratios). I am therefore unsure that your “across species” analyses (figure 1, table 2) are valid. Another concern I have with the “across species” analyses is that you detect an effect of species identity using ANCOVA and don’t consider interactions between species identity and body size. But the graphs clearly show that species identity and body size interact (i.e., different slopes over body size for the different species). If you only present species-specific analyses, your story about body size may disappear or be diminished. That would nevertheless be interesting, as it would suggest that the body-size principle of MTE applies more strongly across species than among individuals within a species. In your mixed-effects models, please address the possibility that you have inflated your sample size for all variables besides ‘time’. Each decapod (and all the predictors that go with it, like its body size and body nutrient composition) will be present as two rows in your data table because you have two time points (30 and 60 minutes). You might just state that you did some preliminary analyses showing higher excretion values at 30 minutes, and then proceed with only analyzing the 30-minute data. I have great admiration for people who write outside of their native language. It is quite remarkable. Your English grammar is, for the most part, fine. However, your word tense or vocabulary choice is sometimes a bit off, so I suggest getting some help with that. Finally, please acknowledge your funding source and ensure that your data are deposited in a publicly available resource. If you can adequately address my comments here and those of the reviewers, and revise your manuscript accordingly, it could be reconsidered.

We look forward to receiving your revised manuscript.

Kind regards,

David B. Lewis, Ph.D.

Academic Editor

PLOS ONE

**Journal requirements:**

“Fondo para la Investigación Científica y Tecnológica, Grant/Award Number: PICT 2016-2542 and PICT 2018-00690 funded this study. CONICET supported the postdoctoral fellowship of G.M. and M.V.T.”

Reviewers' comments:

Reviewer's Responses to Questions

**Comments to the Author**

1. Is the manuscript technically sound, and do the data support the conclusions?

Reviewer #1: Partly

Reviewer #2: No

2. Has the statistical analysis been performed appropriately and rigorously? 

Reviewer #1: I Don't Know

Reviewer #2: I Don't Know

3. Have the authors made all data underlying the findings in their manuscript fully available?

Reviewer #1: Yes

Reviewer #2: Yes

4. Is the manuscript presented in an intelligible fashion and written in standard English?

Reviewer #1: No

Reviewer #2: No

5. Review Comments to the Author

Reviewer #1: I have reviewed the manuscript titled "Consumer-driven nutrient recycling of freshwater decapods: linking ecological theories and application in integrated multi-trophic aquaculture. The questions, goals, and objectives and hypotheses identified for the study and laid out in the manuscript are scientifically well justified, however, as a consumer of the information, overall I found the manuscript, especially the description and presentation of the statistical findings, lengthy and hard to follow.

Overall, I wonder if the authors tried to do to much in a single manuscript and could have been better served by breaking the manuscript into at least two. There were time, especially for the results and text associated with Tables 2-3 that either I didn't understand this statistical approach, or there were inconsistencies in text from the tables.

Second, I found that the overall manuscript was very long. I am not sure if this was due to what all was being covered in the results, or the authors writing style.

Third, I found that the manuscript might benefit from some more traditional word uses (e.g. "conserved" versus "preserved" line 226), some awkward sentences, correcting the flipping back and forth between decapod and crustaceans (I would use decapods for broad experimental group, unless you mean to refer to crustaceans in genera), replacement of pronouns and ambiguous terms like this, it, etc., and because of the use of numbered citations, avoid using the inference of authors names and use more active sentences (like line 86 " For example, [25] found....), and the removal of unnecessary words or sentences.

Fourth, would it make sense to report the PCA first in the results as it is an exploratory data analysis tool, or did you do that at the end based on the results of the univariate stats?

Fifth, would the start of the discussion benefit by a paragraph of identifying the 3-5 main findings of the study to help focus the reader and simply refer to how the finding supported or did not support your expectations.

Reviewer #2: In the present study, the authors tried to test the key factors in determining the nitrogen and phosphorus excretion rates and their ratios of three decapod species. Here, I point out some major concerns on the methods and structure of the ms.

1. The introduction part is too long, and there are too many hypothesizes that I cannot catch the main research question in this study. The introduction tells the audiences with the significance of the study, the gap between the published works and your study, a clear objective of the research, and the research questions. You may get some ideas of scientific writing from the following article:

Kevin W. Plaxco, 2010. The art of writing science. Protein Science, 19:2261—226. DOI: 10.1002/pro.514

2. In the MM part, I was confused if you fed the tested three species with both DF and OF fish-food or fed each species with the selected food? The N:P of the two selected food type was similar (3.45 vs 3.06, calculated from Table 1), so why you decided to use these foods?

For a given species of aquatic consumers, the excretion rates of both N and P, and the excreted N:P were largely determined by the food types (N, P contents and N:P).

What’s the stoichiometric metrics of the three tested species in the present study? Does the N:P varies significantly among them?

The commonly used method for measuring the excretion rates of aquatic animals is directly measure the rate in the lab immediately after collecting the animals from their natural habitats. Did you measure the natural excretion rate of the tested species?

Other minor comments:

Check the format of citations, some have doi links.

Incorrect unit in the Y axis of Fig. 2b.

Reorganize the Result section.

6. PLOS authors have the option to publish the peer review history of their article (what does this mean?). If published, this will include your full peer review and any attached files.

Reviewer #1: No

Reviewer #2: No

---

## [Author Response · Author response to Decision Letter 0]

28 Jul 2022

Response to academic editor

Query 1

The questions and hypotheses were a bit confusingly laid out. In particular, questions 1 and 2 seem redundant. Both questions appear to ask how N, P, and N:P excretion are influenced by a list of several predictor variables. I think you may have distinguished these as two questions to match the two different statistical approaches (mixed-effects models vs. ANCOVA regressions). But ultimately, they seem like the same question to me: how do a bunch of variables (body size, etc.) affect excretion chemistry. Perhaps you can simplify your question statements, followed by a list of predicted relationships (what you currently refer to as hypotheses).

Reply to query 1

We reorganized the questions according to the editor's suggestions and focused on analyzing the effect body size, body composition and feeds on nutrient excretion using only the incubation time of 30 minutes and within taxa. Therefore, we narrow down the first two questions to a single one and eliminate the across taxa analyzes. We kept the list of predicted relationships, as appropriated. 

Query 2

I agree with both authors that, overall, the manuscript was too long. The Discussion alone is nearly 3500 words. Perhaps you can focus the Discussion text on a few key findings. I share reviewer 1’s suspicion that you have included too much. Perhaps consider just presenting the species-specific analyses. There are strong species differences in excretion and body nutrient content and ratios, and the species are different sizes (and size also predicts excretion and body nutrient content and ratios). I am therefore unsure that your “across species” analyses (figure 1, table 2) are valid.

Reply to query 2

We reduced the length of the manuscript (around 2000 words) considering the suggestions of editor and reviewers and tried to focus only in key findings. We decided to eliminate the different scales of analysis (across, among, within taxa), and only performed within and among taxa analyses with the appropriate statistical analysis. We simplified the correlations among variables through collinearity tests and performed the posterior analysis according to these results. It is also important to highlight that we focus the within taxa analyses but, according to the aim of using these decapods in integrated aquaculture systems, it is important to us to carry out the analysis among taxa. These changes simplified the results and thus the overall size of the manuscript.

Query 3

Another concern I have with the “across species” analyses is that you detect an effect of species identity using ANCOVA and don’t consider interactions between species identity and body size. But the graphs clearly show that species identity and body size interact (i.e., different slopes over body size for the different species). If you only present species-specific analyses, your story about body size may disappear or be diminished. That would nevertheless be interesting, as it would suggest that the body-size principle of MTE applies more strongly across species than among individuals within a species.

Reply to query 3

As mentioned before, we eliminated the different scales of analysis and, therefore, the across taxa analyses. We now performed the “among taxa” with appropriate statistical analysis. Firstly, we tested the assumption of ANCOVA (as equality slope and covariate between groups). Because the assumptions were not met (i.e. different body mass of decapod species), we ran ANOVA or Wilcoxon tests. 

Query 4

In your mixed-effects models, please address the possibility that you have inflated your sample size for all variables besides ‘time’. Each decapod (and all the predictors that go with it, like its body size and body nutrient composition) will be present as two rows in your data table because you have two time points (30 and 60 minutes). You might just state that you did some preliminary analyses showing higher excretion values at 30 minutes, and then proceed with only analyzing the 30-minute data.

Reply to query 4 

We performed a preliminary analysis to evaluate the effect of time-lapse incubation (30 and 60 minutes) on the excretion of nutrients using repeated measures ANOVA. We finally selected the 30 minutes data to perform the subsequent analyses.

Query 5

I have great admiration for people who write outside of their native language. It is quite remarkable. Your English grammar is, for the most part, fine. However, your word tense or vocabulary choice is sometimes a bit off, so I suggest getting some help with that.

Reply to query 5

We send the manuscript to revise the vocabulary before the submission to the journal. However, considering the recommendations of editor and reviewer, we remark this comments to the English reviewer, who realized another revision in the new version of the manuscript.

Response to reviewers

Reviewer #1

Query 6

I have reviewed the manuscript titled "Consumer-driven nutrient recycling of freshwater decapods: linking ecological theories and application in integrated multi-trophic aquaculture. The questions, goals, and objectives and hypotheses identified for the study and laid out in the manuscript are scientifically well justified, however, as a consumer of the information, overall I found the manuscript, especially the description and presentation of the statistical findings, lengthy and hard to follow. Overall, I wonder if the authors tried to do to much in a single manuscript and could have been better served by breaking the manuscript into at least two. There were time, especially for the results and text associated with Tables 2-3 that either I didn't understand this statistical approach, or there were inconsistencies in text from the tables. Second, I found that the overall manuscript was very long. I am not sure if this was due to what all was being covered in the results, or the authors writing style.

Reply to query 6

It is true that the original version of the manuscript was long and difficult to follow. Following the recommendations of the editor and reviewers, we decided to omit some analyzes previously carried out (across taxa analyses) and rearranged others (within and among taxa). In addition, we reduce parts of the text to fit the new analysis approach and focus on discussing the most important findings. Regarding Tables 2 and 3, they were deleted because we focused in the within taxa analysis and, the model selection were made after correlation test. This new analysis approach simplified the results and reduced the overall length of the manuscript. 

Query 7

Third, I found that the manuscript might benefit from some more traditional word uses (e.g. "conserved" versus "preserved" line 226), some awkward sentences, correcting the flipping back and forth between decapod and crustaceans (I would use decapods for broad experimental group, unless you mean to refer to crustaceans in genera), replacement of pronouns and ambiguous terms like this, it, etc., and because of the use of numbered citations, avoid using the inference of authors names and use more active sentences (like line 86 " For example, [25] found....), and the removal of unnecessary words or sentences.

Reply to query 7

We followed all recommendations suggested by reviewer 1, both in relation to the use of the words decapods and crustaceans and with respect to the English wording. In the latter case, despite the fact that the manuscript had been previously reviewed, we resubmitted it to an English editing service, highlighting all the suggestions made by the editor and reviewers. 

Query 8

Fourth, would it make sense to report the PCA first in the results as it is an exploratory data analysis tool, or did you do that at the end based on the results of the univariate stats?

Reply to query 8

We decided to use the PCA as a conclusion of the overall results and as a way to facilitate the interpretation by lectors. In addition, the PCA data did not include the types of feed once it was previously discarded in linear models.

Query 9

Fifth, would the start of the discussion benefit by a paragraph of identifying the 3-5 main findings of the study to help focus the reader and simply refer to how the finding supported or did not support your expectations.

Reply to query 9

We omitted the first paragraph of the discussion and replaced it with a new, more concise version of the main findings of the study that we had previously redacted in the subsequent paragraphs.

Reviewer #2

Query 10

The introduction part is too long, and there are too many hypothesizes that I cannot catch the main research question in this study. The introduction tells the audiences with the significance of the study, the gap between the published works and your study, a clear objective of the research, and the research questions. You may get some ideas of scientific writing from the following article:

Kevin W. Plaxco, 2010. The art of writing science. Protein Science, 19:2261—226. DOI: 10.1002/pro.514

Reply to query 10

We appreciate the recommendation of the article. As mentioned previously, we reduced the length of the manuscript, including the introduction, considering the suggestions of editor and reviewers and tried to focus only in key findings. The elimination of the across taxa analysis and the rearrangement of others simplified the results and thus the overall size of the manuscript.

Query 11

In the MM part, I was confused if you fed the tested three species with both DF and OF fish-food or fed each species with the selected food? The N:P of the two selected food type was similar (3.45 vs 3.06, calculated from Table 1), so why you decided to use these foods?

Reply to query 11

We modified the MM text to avoid confusion the following sentence: “Two commercial feeds (Garay SRL) were used and offered separately”. We decided to use these feeds because they are commercial and locally available for the culture of fish with omnivorous and detritivorous trophic habits. This has to do with our goal of using these decapods in integrated aquaculture with fish, which in turn are fed with these feeds. The N:P ratio of theses feeds are calculated using molar values. Therefore, the correct values for this ratio is 17.7 for OF and 11.9 for DF. We decided to include in Table 1 the molar values of nutrient ratios and delete the analysis realized between OF and DF composition. This was due to our aim of evaluate the effect of feed on excretion rates, and not evaluate the differences of feed composition. Type of feed was included as categorical and not as continuous variable in analysis. In addition, the data of feed composition was not enough to infer reliable statistical differences. 

Query 12

For a given species of aquatic consumers, the excretion rates of both N and P, and the excreted N:P were largely determined by the food types (N, P contents and N:P).

What’s the stoichiometric metrics of the three tested species in the present study? Does the N:P varies significantly among them?

Reply to query 12

We interpret that reviewer 2 asks us about the stoichiometry of the body content of decapods. These data are explained in figure 4, in which it is also observed that all decapod species differ among them in terms of body content.

Query 13

The commonly used method for measuring the excretion rates of aquatic animals is directly measure the rate in the lab immediately after collecting the animals from their natural habitats. Did you measure the natural excretion rate of the tested species?

Reply to query 13

We know that this type of approach is commonly used in excretion rates studies. However, our aim was to evaluate it experimentally since in the environment they do not consume artificial feeds. Perhaps in the future we can consider evaluating the excretion rate of these decapods in the field.

Query 14

Other minor comments:

Check the format of citations, some have doi links.

Incorrect unit in the Y axis of Fig. 2b.

Reorganize the Result section.

Reply to query 14

We checked and corrected all the suggestions realized by reviewer 2.

---

## [Decision Letter · Decision Letter 1]

11 Nov 2022

PONE-D-22-00145R1Consumer-driven nutrient recycling of freshwater decapods: linking ecological theories and application in integrated multi-trophic aquaculturePLOS ONE

Dear Dr. Carvalho,

Thank you for submitting your manuscript to PLOS ONE. After careful consideration, we feel that it has merit but does not fully meet PLOS ONE’s publication criteria as it currently stands. Therefore, we invite you to submit a revised version of the manuscript that addresses the points raised during the review process.

Here, I offer several ideas that, in my view, would help the manuscript. Please address all of them with the suggested changes or a strong rationale for why no change is needed.In the within-species analysis, please better describe the statistical modeling approach. It seems that some pre-screening was done with the Spearman correlations before the full models were fit, but I don’t really understand the overall modeling strategy. Please clarify the sequence of steps. Additionally, please clarify why some models (e.g., *A. uruguayana* N excretion) included the diet*body mass interaction while others did not. I think this explanation is in your writing, but I couldn’t understand it.Please consider the constraints and artifacts of your study design. Reviewer 4 makes some important comments (e.g., about stress to the animals) that you should address.Acknowledge the lack of nitrite + nitrate data in your excretion analysis, and the lack of fecal egestion data, and how this may constrain your certainty in your conclusions about organism retention vs. recycling of nutrients. Check units, as reviewer 4 suggests in several comments.Check calculations of stoichiometric molar ratios, as reviewer 4 suggests. I also noticed this problem while reading the paper. For example, consider the *M. borellii* data in figure 3. Median N is about 1.4. Median P is about 0.04. So molar N:P should be about 70-80 ([1.4/14]/[0.04/31]). But if you look at panel 3C, molar N:P is displayed as about 1. Also, if *M. borellii* has the highest N excretion and lowest P excretion of the three species, it should have the highest excretion N:P. In summary, please thoroughly review your stoichiometric calculations in your original data, and in all analyses, graphs, and tables in this manuscript.Hypothesis 2a (line 132) contradicts lines 62-64. Please reconcile.In some cases, terms that were significant in the linear model were not significant when evaluated alone (e.g., in regression; lines 306-308, 329-330). Please consider how to reconcile those differences. Alternatively, you might omit the single-factor regressions and adopt the multi-factor linear models as your definitive results.One shining result in your paper is the PCA. Despite the mixed messages from the other analyses, the PCA very nicely shows the three species separating in multivariate space. I suggest you draw on those findings more in your discussion. Additionally, please introduce the PCA in the data analysis section of the Methods.Please take particular note of reviewer 4’s final two comments. There is a large literature that you could use to more fully contextualize your findings.Writing outside your native language is very admirable. It is not something I could do. However, I think your manuscript could improve with editing by a native English speaker. This might clear up some of the uncertainties I noted above in your statistical analysis.Lastly, please indicate how data will be made available upon publication (e.g., in the EDI data repository)

We look forward to receiving your revised manuscript.

Kind regards,

David B. Lewis, Ph.D.

Academic Editor

PLOS ONE

Reviewers' comments:

Reviewer's Responses to Questions

**Comments to the Author**

1. If the authors have adequately addressed your comments raised in a previous round of review and you feel that this manuscript is now acceptable for publication, you may indicate that here to bypass the “Comments to the Author” section, enter your conflict of interest statement in the “Confidential to Editor” section, and submit your "Accept" recommendation.

Reviewer #3: All comments have been addressed

Reviewer #4: (No Response)

2. Is the manuscript technically sound, and do the data support the conclusions?

Reviewer #3: Partly

Reviewer #4: Partly

3. Has the statistical analysis been performed appropriately and rigorously? 

Reviewer #3: No

Reviewer #4: Yes

4. Have the authors made all data underlying the findings in their manuscript fully available?

Reviewer #3: Yes

Reviewer #4: Yes

5. Is the manuscript presented in an intelligible fashion and written in standard English?

Reviewer #3: No

Reviewer #4: Yes

6. Review Comments to the Author

Reviewer #3: The authors are writing in their non-native language and have sought the help of an English editor. For the most part I was able to understand the paper. There were sections that had incorrect uses of certain words or the meaning was not clear. I think the authors should be given another chance to edit this as there was improvement from the first version and penalizing them for errors of this sort is not appropriate.

Reviewer #4: This study is based on the results of time series (30 and 60 min) incubations of 3 species of decapods preincubated for 2 weeks with a diet from 2 different fish feeds. During incubations dissolved inorganic nitrogen (ammonium) and phosphorus were measured. C, N and P in fish feeds and decapods body mass and elemental composition were also measured. The motivation of the study was to analyze comparatively the elemental composition, the excretion rates and the drivers of excretion and stoichiometry (e.g. body mass, diet) in 3 decapods species potetially relevant in integrated aquacolture.

I was not reviewer during the first round of revision of this manuscript but I went throught the corrections and answers and I acknowledge the work done by authors. However I have a number of doubts about this paper, that I shortly list below.

- I did not catch from the manuscript the number of organisms incubated in the different treatments and a general experimental scheme. The experiment itself is not complex, but schemes help.

- Nitrate and nitrite are also inorganic nitrogen forms. I know they are not metabolic wastes but this is not a reason to avoid their measurements. Microbes growing inside or outside decapods can include nitrifiers, oxidizing the ammonium excreted. This was likely not considered but should be shortly discussed.

- in Table 1 reported molar C:N:P ratios are (very) different from those I calculated from available data (e.g. molar C:N=gC/12:gN:14). It is possible that I am wrong but I ask authors to clarify how they did the calculations.

-Line 167 Units of conductivity are microS/cm

-The method described in line 195 seems very stressfull for macrofauna. In a time-serie experiment I do not understand the need for washing animals with distilled water: why not to perform a simple transfer of decapods to microcosms without fish feed containing the same pre-incubation water and start measurements, after a T0?

-Lines 283-285: I guess correct units are mg ind-1

-Line 279-282: Please check calculations in Table 1

-Figs. 1 and 3: the units of excretion rates should be reported as micrograms N or P per mg dry mass per hour and not per 0.5 h. “half hour interval” is never reported as excretion time unit in the literature.

-Line 366 and Figure 4. Again I tried to calculate molar N:P, C:N and C:P ratios and what is reported in the graphs on the right dos not match with calculations done with the numbers reported in the left column. Please report how these calculations were carried out. As an example the approximate N:P ratio of M. borellii is (10:14)/(1.3/31)=17 but the graph reports a numbler close to 7.5.

- I see in the discussion a very limited (no) comparison of rates and elemental composition with data from the literature, to validate what was measured in this work. Diet is different in nature, but numbers/differences should be comparable. If data are not compared the study is self-referential and local.

-In the whole manuscript feaces as a way to get rid of metabolic waste are not cited, but some of the differences among organisms might be due to release of metabolis waste not via excretion but via feaces. The liteature reports many examples about N and P as dissolved or particulate forms. This needs to be discussed and affects also the entire last paragraph of the discussion. In my opinion this last paragraph exceeds what can be inferred from the reported measurements. There is a vast literature on how organisms as decapods can be integrated in fish aquacolture; here this is explored on the very surface.

7. PLOS authors have the option to publish the peer review history of their article (what does this mean?). If published, this will include your full peer review and any attached files.

Reviewer #3: No

Reviewer #4: No

---

## [Author Response · Author response to Decision Letter 1]

25 Dec 2022

RESPONSE TO REVIEWERS

PONE-D-22-00145R1

Consumer-driven nutrient recycling of freshwater decapods: linking ecological theories and application in integrated multi-trophic aquaculture

PLOS ONE

Response to academic editor

Query 1

In the within-species analysis, please better describe the statistical modeling approach. It seems that some pre-screening was done with the Spearman correlations before the full models were fit, but I don’t really understand the overall modeling strategy. Please clarify the sequence of steps. Additionally, please clarify why some models (e.g., A. uruguayana N excretion) included the diet*body mass interaction while others did not. I think this explanation is in your writing, but I couldn’t understand it.

Response to query 1

We described in more detail the statistical modeling approach. Anyway, when we checked the Spearman correlations, we realized that it was not the better or the only way to selected variables to then run definitive models. We are very grateful for this suggestion because it allowed us to improve the analysis methodology. Initially, we run correlations test to verify the correlations between independent variables (body mass and body elemental content) and not for select the better model. Regarding the pair of variables that presented collinearities and significance, only one variable of each pair was selected as an independent variable for posterior analysis. In addition, we want to notify that depending on the assumptions of normality and homoscedasticity of the variables, we run Spearman o Pearson correlations. For select the best models to respond the variation of excretion rates, we considered that Akaike information criterion (AIC) was the most appropriate method. Therefore, after testing for collinearities, we chosen the linear models for each species based on AIC, starting (for all species) with the interaction between type of feed and body mass, the interaction between type of feed and body elemental content, and the interaction between body elemental content and body mass (~Feed*Body mass + Feed*Body content + Body content*Body mass). Then, we run the selected models to test the significance of the variables. Moreover, we worked with dummies variables. 

When the general model of excretion rate showed many NA (not available) data, we could not run the model with AIC. Then, we separate the general model in small models with the possible combinations of independent interactions and run separately.

Query 2

Please consider the constraints and artifacts of your study design. Reviewer 4 makes some important comments (e.g., about stress to the animals) that you should address.

Response to Query 2

Please, verify the query 16.

Query 3

Acknowledge the lack of nitrite + nitrate data in your excretion analysis, and the lack of fecal egestion data, and how this may constrain your certainty in your conclusions about organism retention vs. recycling of nutrients.

Response to query 3

Please, verify the query 13 

Query 4

Check units, as reviewer 4 suggests in several comments.

Response to query 4

We reviewed and correct all units suggested by reviewer 4. Please check the queries 15, 17 and 19.

Query 5

Check calculations of stoichiometric molar ratios, as reviewer 4 suggests. I also noticed this problem while reading the paper. For example, consider the M. borellii data in figure 3. Median N is about 1.4. Median P is about 0.04. So molar N:P should be about 70-80 ([1.4/14]/[0.04/31]). But if you look at panel 3C, molar N:P is displayed as about 1. Also, if M. borellii has the highest N excretion and lowest P excretion of the three species, it should have the highest excretion N:P. In summary, please thoroughly review your stoichiometric calculations in your original data, and in all analyses, graphs, and tables in this manuscript.

Response to query 5

We thoroughly reviewed our calculations in the original data and found some mistakes. We noticed that the calculations of all figures with ratio values were previously made in µg, not in molar (Figure 2a, b, Figure 3c, and 4 d, e, f). In addition, we log-10 transformed the values of excretion rates among taxa (Figure 3) due to the data scale. Initially, the statistical calculations were made with the correct values. In addition, we found that N:P in the original data sheet had mistaken values due to “copy and taste” errors. These values were corrected and know represent the adequate ratios of nutrient excretion on N:P in M. borellii. 

Query 6

Hypothesis 2a (line 132) contradicts lines 62-64. Please reconcile.

Response to query 6

We corrected line 63. Where we said that body P content should be less in small-bodied organisms, should say that body P content should be high in small-bodied organisms 

Query 7

In some cases, terms that were significant in the linear model were not significant when evaluated alone (e.g., in regression; lines 306-308, 329-330). Please consider how to reconcile those differences. Alternatively, you might omit the single-factor regressions and adopt the multi-factor linear models as your definitive results.

Response to query 7

We decided to omit the single-factor regressions and adopt the multi-factor linear models as definitive results. We only run linear models (without posterior analysis, as single-factor regressions and ancova) because they are more robust to evaluate a group of variables together, and select those with more predictive power. This decision changed partially the results of within taxa analysis and, consequently, we had to modify partially the text in the results and discussion sections.

Query 8

One shining result in your paper is the PCA. Despite the mixed messages from the other analyses, the PCA very nicely shows the three species separating in multivariate space. I suggest you draw on those findings more in your discussion. Additionally, please introduce the PCA in the data analysis section of the Methods.

Response to query 8

We introduced the PCA in the data analysis section of the Methods. In accordance with this suggestion, we wrote in the second paragraph of the discussion, and when appropriate, the main findings of PCA with respect the three species separation in multivariate space. 

Query 9

Please take particular note of reviewer 4’s final two comments. There is a large literature that you could use to more fully contextualize your findings.

Response to query 9

Please, verify the query 22.

Query 10

Writing outside your native language is very admirable. It is not something I could do. However, I think your manuscript could improve with editing by a native English speaker. This might clear up some of the uncertainties I noted above in your statistical analysis.

Response to query 10

Certainly, writing in the non-native language is a challenge but we tried to do our best in each written sentence. If there are words or terms that raise doubts about their meaning, we greatly appreciate that you or the reviewers highlight them, so it is easier, for the English editor and us, to correct those errors. Every time we send the manuscript to review the English, we have a cost to pay, in addition to the time needed, that differs if it is the entire text or a part of it. We appreciate you considering it. If the manuscript now get to a point where acceptance seems more probable, we expected that the editor could lend a hand with finding the unclear sections. With respect the statistical analysis, the uncertainties you noticed probably were clarified with the new statistical description in the methodology (AIC included and Multifactor Linear Models as definitive results). 

Query 11

Lastly, please indicate how data will be made available upon publication (e.g., in the EDI data repository)

Response to query 11

Data will be available at CONICET data repository (https://ri.conicet.gov.ar/).

Response to reviewers

Query 12

I did not catch from the manuscript the number of organisms incubated in the different treatments and a general experimental scheme. The experiment itself is not complex, but schemes help.

Response to query 12

We decided not to carry out a scheme because, as the reviewer commented, the experimental design is not complex. Furthermore, we do not want to increase the length of the manuscript with information that can be well interpreted from the text. However, we failed to clarify the number of individuals of each species that consumed each of the two feeds offered. For this reason, we made a small modification in the text in order to improve the understanding of the experimental design. The change can be visualized in 190 and 191 of the document without track change.

Query 13

Nitrate and nitrite are also inorganic nitrogen forms. I know they are not metabolic wastes but this is not a reason to avoid their measurements. Microbes growing inside or outside decapods can include nitrifiers, oxidizing the ammonium excreted. This was likely not considered but should be shortly discussed.

Response to query 13

Before initiated the trial, we washed all chambers with bleach (reducing the bacteria load) before the test and filled it with filtered water from the same source. We now included in Mat and Met this clarification to emphasize the rigorous care with excretion chambers conditions previously to the trials began. In line 198 we wrote: “Then, each organism was removed from the plastic recipients, washed with distilled water, and transferred to glass bottles (previously cleaned with bleach to reduce the bacteria load) with 150 ml of filtered (MG-F 0.7 µm, Munktell Filter-Sweden) and dechlorinated tap water.” This standardization reduces the variability that may occur due to the nitrifying activity that could occur in 30 minutes. In addition, the measurement of this oxidized N- compounds is not a common methodology observed in the vast existing literature on the same subject. Mcmanamay et al. (2011) is one of the few authors that measured nitrate in the excretion bags, in case of rapid nitrification, and observed that N and P concentration did not change significantly during the incubation time. In this context, we added a brief discussion about this issue in lines 454-457. “In addition, because we did not measured nitrates, 30 minutes of incubation plus standardization of methodology could reduce the error of estimating N excretion due to the action of nitrifying bacteria that oxidize ammonia to nitrites and nitrates [15].”

Query 14

In Table 1 reported molar C:N:P ratios are (very) different from those I calculated from available data (e.g. molar C:N=gC/12:gN:14). It is possible that I am wrong but I ask authors to clarify how they did the calculations.

Response to query 14

Thank you very much for reviewing the values in the table. With this comment we were able to verify that there was an error in the calculations that was due to the fact that the value of g/mol was being multiplied by the value obtained (g/100g) of each element, and not divided, as it should be. The results of Table 1 are now correct.

Query 15

Line 167 Units of conductivity are microS/cm

Response to query 15

We corrected the unity of conductivity

Query 16 

The method described in line 195 seems very stressfull for macrofauna. In a time-serie experiment I do not understand the need for washing animals with distilled water: why not to perform a simple transfer of decapods to microcosms without fish feed containing the same pre-incubation water and start measurements, after a T0?

Response to query 16

Rinsing with distilled water was performed very quickly using a sink while the animal was transferred from one container to another. We decided to carry out this procedure to avoid carrying food remains and other compounds that could be in the pre-incubation chamber. We believe that a “shower” of distilled water has not exacerbated the stress that animals suffer due to the experimental manipulation. In addition, we acclimated the individuals to the experimental conditions, which includes the manipulation during feeding time. In any case, the same procedure was performed with all the specimens, which makes the results obtained with similar error and, therefore, comparable.

Query 17

Lines 283-285: I guess correct units are mg ind-1

Response to query 17

We corrected the units of body mass.

Query 18

Line 279-282: Please check calculations in Table 1

Response to query 18

We checked and corrected the calculations in Table 1. In lines285 and 286, we corrected the text once C:P was not similar between OF and DF.

Query 19

Figs. 1 and 3: the units of excretion rates should be reported as micrograms N or P per mg dry mass per hour and not per 0.5 h. “half hour interval” is never reported as excretion time unit in the literature.

Response to query 19

This suggestion was considered and the excretion rates values were expressed by hour in the figures, but the measures were taken at 30 minutes (this clarification was included in the legends).

Query 20

Line 366 and Figure 4. Again I tried to calculate molar N:P, C:N and C:P ratios and what is reported in the graphs on the right dos not match with calculations done with the numbers reported in the left column. Please report how these calculations were carried out. As an example the approximate N:P ratio of M. borellii is (10:14)/(1.3/31)=17 but the graph reports a numbler close to 7.5.

Response to query 20

Again, thank you very much for reviewing the molar values on the graphs. What happened is that at the time of graphing the results we used the data of % (g/100g) and not of molar. We correct the graphs with the ratio in molar unit.

Query 21

I see in the discussion a very limited (no) comparison of rates and elemental composition with data from the literature, to validate what was measured in this work. Diet is different in nature, but numbers/differences should be comparable. If data are not compared the study is self-referential and local.

Response to query 21

We added to the discussion (Nutrient excretion - among taxa) a paragraph about the difficult to compare nutrient excretion rates among different species, methodologies, and unities of measure (lines 547-563). We also added relevant data about body content of other prawn species (we did not found information about others anomurans and crbas) (lines 593-600).

Query 22

In the whole manuscript feaces as a way to get rid of metabolic waste are not cited, but some of the differences among organisms might be due to release of metabolis waste not via excretion but via feaces. The liteature reports many examples about N and P as dissolved or particulate forms. This needs to be discussed and affects also the entire last paragraph of the discussion. In my opinion this last paragraph exceeds what can be inferred from the reported measurements. There is a vast literature on how organisms as decapods can be integrated in fish aquacolture; here this is explored on the very surface.

Response to query 22

We did take into account the nutrients eliminated through egestion and we included it in the discussion previously (line 534 and 647). Of course the literature reports many examples about N and P as dissolved or particulate forms, but our results are focused on the excretion/mineralisation of inorganic nutrients and it would be to extend the discussion towards topics that exceed the objectives of the present work. However, to attain the reviewer's comment, we decided to expand briefly the discussion regarding this topic (lines 488-492).

The last paragraph of the discussion is a brief conclusion of the main findings and what studies are needed to further understand the role of this decapods as animal-mediated nutrient cycling. Considering this, we do not understand why, in your opinion, the discussion of nutrient egestion (via faeces) affects the entire last paragraph. 

However, we believe that at some point the reviewer refers to the last item of the discussion (Comments about application in IMTA), not the last paragraph. In this section we added bibliography of recent studies (among the few available) that refer to the culture of a prawn of the genus Macrobrachium (as non-fed species, not the main cultured species) with other fish species in an IMTA context (lines 613-625). The existing information is much broader for the species of crustaceans that are fed (main cultured species), that is, those that do not act as waste recyclers.

---

## [Editor Report · Decision Letter 2]

5 Feb 2023

PONE-D-22-00145R2Consumer-driven nutrient recycling of freshwater decapods: linking ecological theories and application in integrated multi-trophic aquaculture

PLOS ONE

Dear Dr. Carvalho,

Thank you for submitting your manuscript to PLOS ONE. After careful consideration, we feel that it has merit but does not fully meet PLOS ONE’s publication criteria as it currently stands. Therefore, we invite you to submit a revised version of the manuscript that addresses the points raised during the review process.

In light of the review comments, you provided thoughtful replies and meaningful revisions to the manuscript. I think the manuscript is improved. I know you have worked hard to provide two revisions, although I think there are some more changes you can make, or clarifications you can provide, to improve it. Please respond to the following questions and suggestions. Even though there are many questions and suggestions below, they all seem pretty straightforward, so I am submitting a recommendation of ‘minor revisions’.

1.
L 22: Change “allometrically within taxa” to “allometrically with body mass” (which I think you meant) 2.
L 25: Change sentence to read: “Feed interacted with body mass to explain P excretion…” 3.
L 29: Delete “by” 4.
L 96: Change “resembles” to “reflects” 5.
L 97: Change “they could” to “their diet could” 6.
L 97: Change “in addition” to “and in response” 7.
L 101: Change “wide” to “substantial” 8.
L 111: Change “potentiality” to “potential” 9.
L 124: Change “of” to “as” 10.
Are hypotheses 1a and 1b meaningfully different? They both seem to predict that N and P excretion correlate with body mass. Perhaps you could delete one of them. 11.
In hypothesis 3a, you propose that body C:N decreases in lower trophic positioned species. This seems to contradict the sentence on lines 82-84, in which you imply that C:N decreases at higher (predator) trophic positions. Please resolve or explain this seeming discrepancy. 12.
L 136 (Hyp 3a): This hypothesis is unclear with regard to N:P. Is N:P going up or down in carcinized decapods and/or at lower trophic positions? 13.
Table 1: Remove “%” from the header. You already provide g/100g as units. 14.
L 192-3: “arranged randomly and interspecified…” – I do not know what you mean by “interspecified”. I think you can just delete “and interspecified”.  15.
L 241: Change “analyses” to “analyze” 16.
L 243: Change “as dummy variable” to “as an indicator variable”. And then state which feed type was 1 and which was 0. Knowing which was 1 and 0 will help readers interpret effects in the model. 17.
L 244: I understand that you evaluated collinearity among body mass and body elemental content because they were both predictor variables and you didn’t want to include two collinear variables as predictors in the same model. However, I do not understand why you examined collinearity among N, P and N:P excretion. These are response variables, all in separate models, so collinearity among them doesn’t really matter. 18.
General comment about modeling. You only present the “final” model in Table 2. I assume you had some full model (e.g., feed, body mass, body element content, all possible 2-term interactions), and then you fit all possible subsets of that full model to find the one with the lowest AIC. Is this correct? If so, please explain that. Also, in supplemental information, please provide the results of all model subset fits with some diagnostic criteria (e.g., AIC, R2), perhaps as a table. 19.
L 274-5: You state that you log-transformed variables to meet assumptions of normality. This seems to slightly contradict the previous paragraph (L 269-272), which implies that sometimes the assumptions of normality were not met, and so non-parametric statistics were used (in which case, I assume you would not log transform, as there would be no reason to in rank-based non-parametric analyses). Do you really mean that you log-transformed, and if the assumptions of normality were still not met after transforming, then you used non-parametric stats? 20.
L 297: Change “more N at” to “N faster over” 21.
L 298: Change “at 60” to “over 60” 22.
L 309: At end of line, change “any” to “no” 23.
L 311-312: On L 311 you say there was collinearity (implying a significant relationship), but on L 312, you note p > 0.05 24.
L 329-330: These results are an example of when it would be helpful to know which feed type =1 and which type =0 in your indicator (“dummy”) variable 25.
L 332: Why were some data NA for N:P? Was it because P = 0 (i.e., below detection limit), so the ratio can’t be calculated? 26.
L 334: “run separately” – Do you mean that you fit several single-factor models of N:P vs. one predictor variable? If so, please clarify. 27.
L 335: Change “differences were showed” to “effects were identified” 28.
Table 2: Include actual coefficient values, not just the sign. This way, readers can identify the magnitude by which an interaction term affects the slope of a factor. 29.
Table 2 – A. uruguayana – Excretion N: Change “DF*Body mass” to “Feed*Body mass” 30.
L 347-349: It appears that you’re reporting whether body element content and stoichiometric ratios were correlated with body mass. But I do not understand why you say there were no “significant linear relations and neither collinearities”. Why make reference to both linear relations and collinearities? Are you reporting two different groups of analyses here? 31.
Please start your Discussion with a straightforward “take-home” message about whether each hypothesis was supported, and if so, by which species or nutrient. This will help your readers link your findings back to your hypotheses. You could even organize this as a table if that helps you. 32.
L 462: Change “MET” to “MTE” 33.
L 472: Change “found” to “find” 34.
L 483: Delete “were” 35.
L 484: I do not understand the references to negative and positive allometry when discussing effects of feed type. 36.
L 493: Add “to” after “respect” (“With respect to…”) 37.
L 494: Change “to” to “for” 38.
L 497: Add “to” after “feasible” (“feasible to collect…”) 39.
L 522-523: Rewrite as “…main factors separating the species in multivariate space.” 40.
L 523: Change “was” to “were” 41.
L 549: Change “unities” to “units” 42.
L 556: Make the “4” a subscript in “NH4-N” 43.
L 560: Change “Despite” to “Although” 44.
L 560: Change “unity” to “units” 45.
L 562: Change “unities” to “units” 46.
L 566: Delete “due to the” 47.
L 566: Change “content” to “contents” 48.
Be sure to upload high quality figures. They are a little fuzzy in my PDF copy

We look forward to receiving your revised manuscript.

Kind regards,

David B. Lewis, Ph.D.

Academic Editor

PLOS ONE
---

## [Author Response · Author response to Decision Letter 2]

28 Feb 2023

1. L 22: Change “allometrically within taxa” to “allometrically with body mass” (which I think you meant)

We changed the text to “allometrically with body mass within taxa”.

2. L 25: Change sentence to read: “Feed interacted with body mass to explain P excretion…”

 We changed the text to “Feed interacted with body mass to explain P excretion of anomurans and N ex-cretion of crabs”.

3. L 29: Delete “by”

We deleted it.

4. L 96: Change “resembles” to “reflects”

We changed it.

5. L 97: Change “they could” to “their diet could”

We changed it.

6. L 97: Change “in addition” to “and in response”

We changed it.

7. L 101: Change “wide” to “substantial”

We changed it.

8. L 111: Change “potentiality” to “potential”

We changed it.

9. L 124: Change “of” to “as”

We changed it.

10. Are hypotheses 1a and 1b meaningfully different? They both seem to predict that N and P excretion correlate with body mass. Perhaps you could delete one of them.

It is true that they seem very similar. However, while hypothesis 1a affirms that body mass is the variable that best explain nutrient excretion, 1b should state the slope of this relation (negative allometry accord-ing to literature). Therefore, we rewrote this hypothesis as follow: “N and P excretion rates negatively scale with body mass”.

11. In hypothesis 3a, you propose that body C:N decreases in lower trophic positioned species. This seems to contradict the sentence on lines 82-84, in which you imply that C:N decreases at higher (preda-tor) trophic positions. Please resolve or explain this seeming discrepancy.

We resolved the discrepancy and changed the text to “Body C:N content increases in carcinized decapods and decreases in higher trophic positioned species”

12. L 136 (Hyp 3a): This hypothesis is unclear with regard to N:P. Is N:P going up or down in carcinized decapods and/or at lower trophic positions?

 We clarified this question rewriting the sentence to “Body C:N content increases in carcinized decapods and decreases in higher trophic positioned species, while N:P increases in more carnivorous species”

13. Table 1: Remove “%” from the header. You already provide g/100g as units.

 We removed it.

14. L 192-3: “arranged randomly and interspecified…” – I do not know what you mean by “interspeci-fied”. I think you can just delete “and interspecified”. 

 We deleted it.

15. L 241: Change “analyses” to “analyze”

We changed it.

16. L 243: Change “as dummy variable” to “as an indicator variable”. And then state which feed type was 1 and which was 0. Knowing which was 1 and 0 will help readers interpret effects in the model.

This was added and corrected. Also, it was indicated in the table 2 as “DF” (feed type 1).

17. L 244: I understand that you evaluated collinearity among body mass and body elemental content because they were both predictor variables and you didn’t want to include two collinear variables as predictors in the same model. However, I do not understand why you examined collinearity among N, P and N:P excretion. These are response variables, all in separate models, so collinearity among them doesn’t really matter.

The collinearity among N, P and N:P excretion was deleted to the sentence. 

18. General comment about modeling. You only present the “final” model in Table 2. I assume you had some full model (e.g., feed, body mass, body element content, all possible 2-term interactions), and then you fit all possible subsets of that full model to find the one with the lowest AIC. Is this correct? If so, please explain that. 

This was better explained in the manuscript. Also, we forget clarify that we used stepAIC function of the MASS package to selected the best fit models. But this was included now. 

Also, in supplemental information, please provide the results of all model subset fits with some diagnos-tic criteria (e.g., AIC, R2), perhaps as a table.

We provided the results of AIC of all model subset using stepAIC function in a table as supplemental in-formation.

19. L 274-5: You state that you log-transformed variables to meet assumptions of normality. This seems to slightly contradict the previous paragraph (L 269-272), which implies that sometimes the assumptions of normality were not met, and so non-parametric statistics were used (in which case, I assume you would not log transform, as there would be no reason to in rank-based non-parametric analyses). Do you really mean that you log-transformed, and if the assumptions of normality were still not met after transforming, then you used non-parametric stats?

This was clarified in each paragraph of this section of the manuscript: In within and among taxa analysis, the variables used were log10-transformed to meet assumptions of normality and heterogeneity of vari-ances. In addition, for working with linear relationships between quantitative variables, normality and homoscedasticity assumptions were tested on the models. In within taxa analysis, if the assumptions of normality and homoscedasticity were still not met after transforming, data without log10-transformed were used in non-parametric stats. Wilcoxon pairwise test was run if the assumptions of normality and homoscedasticity were not met in ANCOVA model. If these assumptions were met, but ANCOVA assump-tions was not met (different slopes and covariate), one-way ANOVA with Tukey pairwise was run.

20. L 297: Change “more N at” to “N faster over”

We changed it.

 21. L 298: Change “at 60” to “over 60”

We changed it.

22. L 309: At end of line, change “any” to “no”

We changed it.

23. L 311-312: On L 311 you say there was collinearity (implying a significant relationship), but on L 312, you note p > 0.05

We corrected it.

24. L 329-330: These results are an example of when it would be helpful to know which feed type =1 and which type =0 in your indicator (“dummy”) variable

We clarified it in data analysis.

25. L 332: Why were some data NA for N:P? Was it because P = 0 (i.e., below detection limit), so the ratio can’t be calculated?

What happens is that there is little data of the N:P body content of T. borellianus. We had a technical problem when we sent the samples to determine elemental composition. The required gas used in the atmosphere of the elemental analyser ran out, which implied that the analyzer returned unrealistic val-ues for N in several samples and we lost samples. The amount of samples that we had was small (200 mg of sample is required for a duplicate determination) and therefore we could not repeat the analyses. So, some individuals remained with P values but not with N values and, for this reason, we calculated N:P values only in few individuals. Then, when we run the full model with several interactions, the number of NA data reduced the freedom degrees with respect the number of variables and the model failed. This was clarified in the manuscript. 

26. L 334: “run separately” – Do you mean that you fit several single-factor models of N:P vs. one predic-tor variable? If so, please clarify.

We mean that several models of N:P vs. each interaction predictor variable were run (N:P vs. Feed*Body mas; N:P vs. N:P body*Body mass; N:P vs. Feed*N:P body). We clarified this in the manuscript. In this way, the independent variables that interact are reduced and therefore the degrees of freedom.

27. L 335: Change “differences were showed” to “effects were identified”

We changed it. 

28. Table 2: Include actual coefficient values, not just the sign. This way, readers can identify the magni-tude by which an interaction term affects the slope of a factor. 

This was included in the table 2. Some of the statistical values changed but the relationships did not. This is because we had to double the values for excretion rate in 30 minutes (following the suggestion of one reviewer) so that it can be expressed per hour and not half an hour.

29. Table 2 – A. uruguayana – Excretion N: Change “DF*Body mass” to “Feed*Body mass”.

We changed it. 

30. L 347-349: It appears that you’re reporting whether body element content and stoichiometric ratios were correlated with body mass. But I do not understand why you say there were no “significant linear relations and neither collinearities”. Why make reference to both linear relations and collinearities? Are you reporting two different groups of analyses here?

We reported two different groups of analyses and this was included and clarified in data analysis section.

31. Please start your Discussion with a straightforward “take-home” message about whether each hy-pothesis was supported, and if so, by which species or nutrient. This will help your readers link your find-ings back to your hypotheses. You could even organize this as a table if that helps you.

We started the discussion with a take-home message followed by answering each hypothesis and predic-tion made in the introduction. This certainly will help readers to link results with hypothesis. 

32. L 462: Change “MET” to “MTE”

We changed it.

33. L 472: Change “found” to “find”

We changed it.

34. L 483: Delete “were”

 We deleted it.

35. L 484: I do not understand the references to negative and positive allometry when discussing effects of feed type.

Negative and positive allometry is referred to the slope of the interaction between body mass and nutri-ent excretion. To avoid confusion, we decided to change the rewrite the test as follows: Regardless of the biomass, anomurans excreted more P and crabs excreted more N when fed OF. When body mass inter-acted with excretion, anomurans had negative allometry for P excretion and crabs positive allometry for N excretion.

36. L 493: Add “to” after “respect” (“With respect to…”)

We added it.

37. L 494: Change “to” to “for”

We changed it.

38. L 497: Add “to” after “feasible” (“feasible to collect…”)

We added it

39. L 522-523: Rewrite as “…main factors separating the species in multivariate space.”

We rewrote it.

40. L 523: Change “was” to “were”

We changed it.

41. L 549: Change “unities” to “units”

We changed it.

42. L 556: Make the “4” a subscript in “NH4-N”

We subscribed it.

43. L 560: Change “Despite” to “Although”

We changed it.

44. L 560: Change “unity” to “units”

We changed it.

45. L 562: Change “unities” to “units”

We changed it.

46. L 566: Delete “due to the”

We deleted it.

47. L 566: Change “content” to “contents”

We changed it.

48. Be sure to upload high quality figures. They are a little fuzzy in my PDF copy

The figures were in good quality when we uploaded it. However, we modified them again according to the dimension specifications (2250 pixels of width maximum and 2625 pixels of height maximum, both at 300 dpi). Before these modifications, figures had lower pixels of width and height but higher dpi (600). Likewise, I think that the pdf converter lowers the quality of the image and that is why it looks fuzzy. The original looks great.

Journal Requirements:

We added a new cite for an statistic procedure. It is with the number 85 - Ripley B, Venables B, Bates DM, Hornik K, Gebhardt A, Firth D, Ripley MB. Package ‘mass’. Cran r, 2013: 538, 113-120 .

---

## [Editor Report · Decision Letter 3]

2 Mar 2023

Consumer-driven nutrient recycling of freshwater decapods: linking ecological theories and application in integrated multi-trophic aquaculture

PONE-D-22-00145R3

Dear Dr. Carvalho,

We’re pleased to inform you that your manuscript has been judged scientifically suitable for publication and will be formally accepted for publication once it meets all outstanding technical requirements.

Kind regards,

David B. Lewis, Ph.D.

Academic Editor

PLOS ONE